# Economic use of plants is key to their naturalization success

Mark van Kleunen [1,2 ✉], Xinyi Xu [3], Qiang Yang [2], Noëlie Maurel [2], Zhijie Zhang [2], Wayne Dawson[4], Franz Essl[5], Holger Kreft[6,7], Jan Pergl [8], Petr Pyšek[8,9], Patrick Weigelt[6], Dietmar Moser[5], Bernd Lenzner [5] & Trevor S. Fristoe [2]

Humans cultivate thousands of economic plants (i.e. plants with economic value) outside their native ranges. To analyze how this contributes to naturalization success, we combine global databases on economic uses and naturalization success of the world's seed plants. Here we show that naturalization likelihood is 18 times higher for economic than non-economic plants. Naturalization success is highest for plants grown as animal food or for environmental uses (e.g. ornamentals), and increases with number of uses. Taxa from the Northern Hemisphere are disproportionately over-represented among economic plants, and economic plants from Asia have the greatest naturalization success. In regional naturalized floras, the percentage of economic plants exceeds the global percentage and increases towards the equator. Phylogenetic patterns in the naturalized flora partly result from phy-logenetic patterns in the plants we cultivate. Our study illustrates that accounting for the intentional introduction of economic plants is key to unravelling drivers of plant naturalization.

[1] Zhejiang Provincial Key Laboratory of Plant Evolutionary Ecology and Conservation, Taizhou University, Taizhou 318000, China. [2] Ecology, Department of Biology, University of Konstanz, Universitätsstrasse 10, D-78457 Konstanz, Germany. [3] Ministry of Education Key Laboratory for Biodiversity Science and Ecological Engineering, Institute of Biodiversity Sciences, School of Life Sciences, Fudan University, Shanghai 200433, China. [4] Department of Biosciences, Durham University, South Road, Durham DH1 3LE, UK. [5] Division of Conservation Biology, Vegetation Biology and Landscape Ecology, Department of Botany and Biodiversity Research, University of Vienna, Rennweg 14, 1030 Vienna, Austria. [6] Biodiversity, Macroecology & Biogeography, University of Goettingen, Büsgenweg 1, D-37077 Göttingen, Germany. [7] Centre of Biodiversity and Sustainable Land Use (CBL), University of Goettingen, Büsgenweg 1, D-37077 Göttingen, Germany. [8] Czech Academy of Sciences, Institute of Botany, CZ-252 43 Průhonice, Czech Republic. [9] Department of Ecology, Faculty of Science, Charles University, Viničná 7, CZ-128 44 Prague, Czech Republic. ✉email: mark.vankleunen@uni-konstanz.de

With human assistance, thousands of organisms have established self-sustaining wild populations beyond their native ranges[1]. These so-called naturalized aliens are a defining characteristic of the distribution of the Earth's biota in the Anthropocene[2], and their numbers are rapidly increasing[3,4]. While some alien organisms have been introduced unintentionally (e.g., as hitchhikers in cargo), many introductions are the result of deliberate activities[5,6]. This is especially the case for economic plants, i.e., plants that are useful for humans as, e.g., food, cooking spices, medicines, construction materials, or ornamentation[7,8].

Humans have cultivated alien plants from the Late Pleistocene onwards[9], but the introduction of alien plants of economic value drastically increased since the 15th century, when Europeans started exploring the world. The European royal houses provided explicit instructions to the captains of their fleets to return with plants of potential economic value[10,11]. Professional plant hunters were sent to distant lands in search of new species or varieties that could be cultivated for a variety of economic uses[12,13]. Many botanical gardens in Europe and the European colonies were established for acclimatizing and testing the economic potential of these alien plants[14]. Some countries, such as Australia, New Zealand, and the USA, even established governmental departments for the introduction of economic plants[15–17]. While it is well known that many naturalized plants are escapees from cultivation[18,19], the contribution of economic taxa to the global-scale patterns of naturalization has yet to be assessed.

Major drivers of plant naturalization success are thought to be propagule pressure (i.e., the number and magnitude of release events)[20,21], residence time[22–24], and intrinsic characteristics of the introduced species[25,26]. Economic plants should hence have more opportunities for introduction and spread than most non-cultivated species, as they are actively propagated and repeatedly planted in large numbers over large areas. However, economic plants are likely to vary in their frequency and extent of cultivation (which, in turn, affects propagule pressure) as well as their traits, depending on the type and number of uses they are grown for[27]. For example, in contrast to most species grown for medicinal purposes, species grown for fodder production are sown over large areas[28] and are likely to be fast-growing, high-yield species. These biological characteristics are also frequently associated with naturalized and invasive species[29–32]. Similarly, species that are utilized for many different purposes are likely to be cultivated more extensively than species with only limited or specialized uses. Therefore, the naturalization success is probably not equal among different types of economic uses, and species cultivated for a wider variety of uses could be more likely to naturalize than those with only a single use.

In the few studies that have analyzed the means of introduction for regional naturalized alien floras, cultivation was frequently identified as a main pathway[33–35]. How this generalizes globally remains unknown. The proportion of naturalized species is usually lower in tropical than in temperate regional floras, and is higher on islands than on the mainland[36–38]. In regions that appear to be more resistant to naturalization by alien species, such as those in the mainland tropics, increased propagule pressure (e.g., by frequent cultivation) may be more critical for the successful establishment of alien populations. Economic taxa may therefore be expected to make up a higher proportion of the naturalized flora in these regions compared with temperate regions or on islands. On the other hand, it is also conceivable that on remote oceanic islands, which are usually depauperate in native plants, humans needed to introduce a greater number of plants to satisfy socio-economic needs, and so these economic plants will constitute a larger proportion of the naturalized island flora. Moreover, the economic uses that contribute the most taxa to naturalized floras in different regions remain unknown.

It was recently shown that the continents of the Northern Hemisphere are overrepresented as donors of naturalized plants[39]. Europe specifically has donated almost four times as many species to the rest of the world than expected based on its relatively small native flora[39]. Charles Darwin was the first to suggest that the specific evolutionary histories of Northern Hemisphere continents might have led to highly competitive biotas that are highly capable of invading elsewhere[40,41]. Furthermore, due to the long history of large-scale human disturbances in Europe, there has likely been a strong selection for species that are preadapted to the more recent large-scale human disturbances in other continents[42]. However, the global naturalization success of European plants is at least to some extent due to deliberate introductions of economic plants to the overseas colonies of the European empires[5,43,44]. Whether Europe and the other Northern Hemisphere continents are also overrepresented as donors of naturalized plants with economic uses, however, has never been assessed.

Naturalized plant species are also more frequent in some families (e.g., Pinaceae, Poaceae, and Rosaceae) than in others[37]. While this pattern may be driven by closely related species sharing traits that promote naturalization success, it may also reflect a phylogenetic bias in species introductions. For example, it is likely that economic plants are not a random subset from the phylogenetic tree, as closely related taxa are likely to share properties that make them more or less useful to humans. In addition, some properties preferred in economic plants might also directly promote naturalization, as has been shown for winter hardiness in temperate regions[31]. If economic use is a major driver of naturalization, this could help to explain the relatively clustered phylogenetic distribution of naturalized plants.

Here, we assess how the economic use of seed plants contributes to their global naturalization success by combining the World Economics Plants (WEP) database[45] with the Global Naturalized Alien Flora (GloNAF) database[46]. Our main questions are: (1) is the percentage of taxa that have become naturalized higher among economic plants than among noneconomic plants, and are certain categories of economic use more strongly associated with naturalization success than others? (2) Are the Northern Hemisphere continents, and Europe in particular, disproportionally overrepresented as donors of economic plants, irrespective of their naturalization success? (3) Are economic plants from continents in the Northern Hemisphere more likely to become naturalized? (4) Do tropical regions and islands have larger proportions of economic plants in their naturalized floras? (5) Do phylogenetic biases with regard to economic uses of plants underlie phylogenetic patterns in naturalization success? We show that naturalization likelihood is higher for economic than noneconomic plants, and conclude that accounting for the intentional introduction of economic plants is key to unraveling drivers of plant naturalization.

## Results

**Naturalization success of economic plants.** Of the 326,101 taxa (including species, subspecies, and varieties) of seed plants with accepted names in The Plant List (http://www.theplantlist.org/), 11,685 have known economic uses according to the WEP database, and 12,013 are naturalized somewhere in the world according to the GloNAF database. Whereas 3.7% of the taxa in the global flora are known to be naturalized, the percentage of economic taxa that are naturalized is an order of magnitude higher (41.0%, resampling test: $p < 0.001$; Fig. 1). Alternative tests using a binomial generalized linear model (GLM) and a phylogenetic binomial GLM gave very similar results (Supplementary Fig. 1).

For all 12 economic use categories included in the WEP database (Table 1), the percentages of naturalized taxa are clearly higher than the percentage in the global flora, but there is considerable variation among the categories (Fig. 2a). Resampling tests suggest that the percentage of taxa used as materials (e.g., for the production of essential oils, fibers, and wax) does not differ from the overall percentage of naturalized taxa in WEP, while the taxa used as gene sources for crop improvement include significantly fewer naturalized taxa than expected (Fig. 2a). The other ten categories include significantly higher percentages of naturalized taxa than the overall percentage in WEP, with the highest percentages for bee plants (i.e., for honey production) and non-vertebrate poisons (i.e., for pest control; Fig. 2a). Similarly, for the subset of naturalized economic plants, the number of

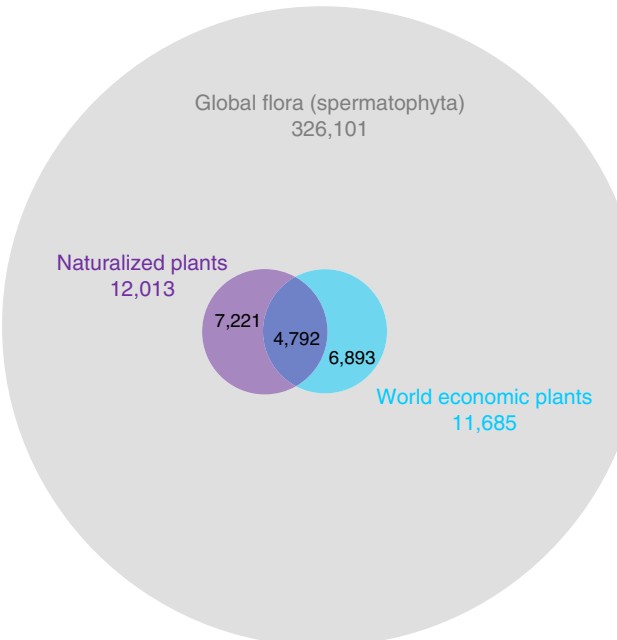

**Fig. 1 The numbers of naturalized and economic plants in the extant global flora.** The size-proportional Venn diagram shows the overlap in the number of taxa between the economic plants of the WEP dataset and the naturalized plants of the GloNAF database. For reference, the gray circle indicates the number of taxa with accepted names in the extant global seed plant flora according to The Plant List (http://www.theplantlist.org/; version 1.1 from September 2013).

regions where a taxon is recorded as naturalized was lowest for gene sources and materials, and highest for bee plants (Supplementary Fig. 2).

Many taxa, however, have multiple economic uses (Supplementary Figs. 3 and 4), which may confound the relationships between naturalization success and each particular economic use. If we only consider taxa with a single kind of economic use, all categories, except the ones with very few taxa (social, invertebrate food, and non-vertebrate poisons), still include higher proportions of naturalized species than observed in the global flora (Fig. 2b). Among those taxa with single economic uses, the percentages of naturalized taxa for plants cultivated as animal food (i.e., as fodder and forage) and for environmental purposes (i.e., plants used for erosion control, soil improvement, agroforestry, and as ornamentals; Table 1) are significantly higher than the overall percentage in WEP, indicating that these economic uses are most strongly associated with naturalization success. Naturalized taxa were significantly less prevalent among plants only used as materials, gene sources, fuels (e.g., firewood and charcoal), for social reasons (e.g., for religious/secular purposes or for their psychoactive properties), as invertebrate food (e.g., as host plants for lac insects or silkworms) or non-vertebrate poisons (Fig. 2b). In alternative tests using a binomial GLM and a phylogenetic binomial GLM of the extant global seed plant flora, plants cultivated as animal food and for environmental purposes still had the highest naturalization probability (Fig. 2c, Supplementary Fig. 5, and Supplementary Tables 1 and 2).

With an increasing number of economic uses, a taxon is more likely to be naturalized (GLM: $z = 29.34$, $p < 0.001$; phylogenetic GLM: $z = 32.09$, $p < 0.001$; $n = 11,685$; Fig. 3a), and, if naturalized, to be so in more regions around the world (Kendall–Theil Sen Siegel nonparametric regression: $V = 7,937,443$, $p < 0.001$; phylogenetic linear model (LM): $t = 22.09$, $p < 0.001$; $n = 4792$; Fig. 3b). To test whether the naturalization probability depends on the specific combination of uses, we analyzed the subset of taxa with no, one or two economic uses with a phylogenetic binomial GLM (Fig. 4 and Supplementary Table 3). If a combination includes an environmental economic use that has a strong positive effect on naturalization as a single use, the probability of naturalization is also relatively high (Fig. 4). On the other hand, when one of the two economic uses is materials, which has a weak positive effect on naturalization as a single use, the proportion of naturalized taxa is in many cases also relatively low (Fig. 4). While the effects of multiple economic uses mainly follow from the main effects of the single uses, the negative

**Table 1 The 12 categories of economic uses from the World Economic Plants database that were included in our analyses.**

| Economic use category | Description | No. of taxa |
|---|---|---|
| Animal food | Fodder and forage for domestic animals | 836 (580) |
| Bee plants | Important honey plants | 202 (149) |
| Environmental | Plants used for erosion control, soil improvement, agroforestry, and as ornamentals | 5666 (3139) |
| Food additives | Plants used as minor constituents of human food preparation (e.g., for flavoring and coloring) | 462 (282) |
| Fuels | Plants used as fuel or grown for biomass production (e.g., for firewood, charcoal, and petroleum substitutes) | 206 (130) |
| Gene sources | Sources of beneficial genes for improvement of crop plants | 2758 (757) |
| Human food | Plants consumed by humans (e.g., fruits and vegetables) | 1360 (757) |
| Invertebrate food | Hosts for beneficial invertebrates (e.g., for lac insects, wax insects and silkworms) | 41 (28) |
| Materials | Plants used for fiber, timber, gums, resins, oils, and wax | 2190 (906) |
| Medicines | Sources of pharmaceutical agents and plants used in folklore medicine | 3104 (1621) |
| Non-vertebrate poisons | Plants used for organic pesticides (e.g., insecticides and herbicides) | 44 (32) |
| Social | Plants with social uses (e.g., for religious/secular purposes, and as stimulants or hallucinogens) | 128 (66) |

For each category the number of taxa is given and in brackets the number of taxa that have become naturalized. The subcategories of each main economic use category can be seen in Supplementary Fig. 11.

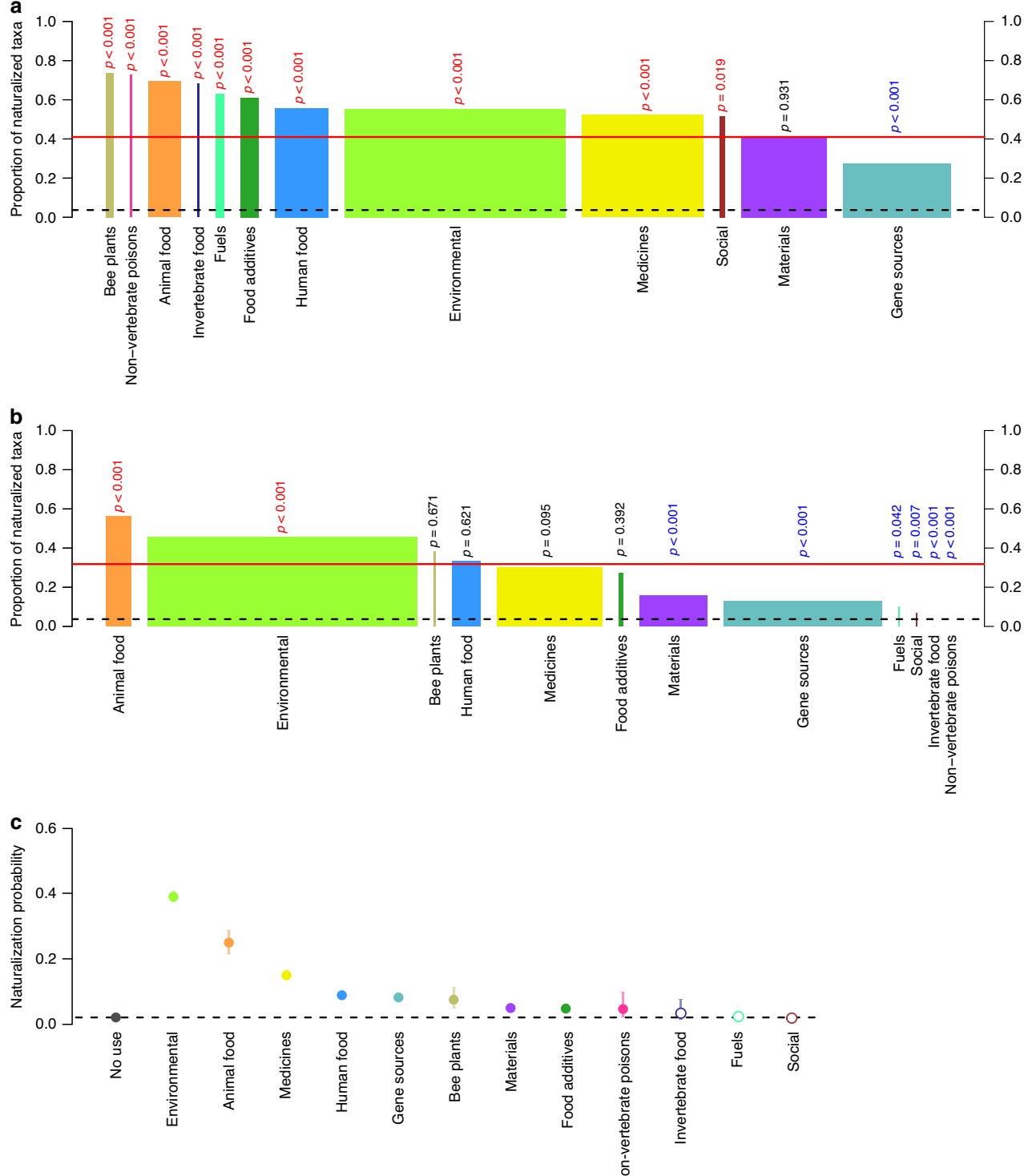

**Fig. 2 The proportion of naturalized taxa within each economic use category.** Spineplots for (**a**) all taxa in the WEP dataset ($n = 11,685$), and (**b**) the subset of taxa with a single economic use ($n = 8246$), and (**c**) naturalization probabilities estimated with a phylogenetic GLM on all seed plants ($n = 326,101$). In (**a**, **b**), the width of each bar is proportional to the number of taxa. The economic use categories are ranked according to (**a**, **b**) the proportion of naturalized taxa or (**c**) naturalization probability. As reference proportions, the red solid lines indicate the proportions of naturalized taxa in (**a**) the full WEP dataset and (**b**) the single-use subset, respectively. The black dashed line indicates (**a**, **b**) the proportion of naturalized taxa in the extant global seed plant flora (economic and noneconomic plants combined) or (**c**) the naturalization probability of taxa without economic use. In (**a**, **b**), $p$ values from two-sided resampling tests indicate whether the proportion of naturalized taxa is significantly higher (red) or lower (blue) than expected or does not deviate from expectations (black). In (**c**), the error bars indicate 95% confidence intervals, and filled circles indicate economic uses where the naturalization probability differs significantly from taxa with no economic uses ($p < 0.05$; for exact $p$ values see Supplementary Table 1). No corrections were made for multiple comparisons. Supplementary Fig. 11 shows spineplots for the subcategories within each economic use category.

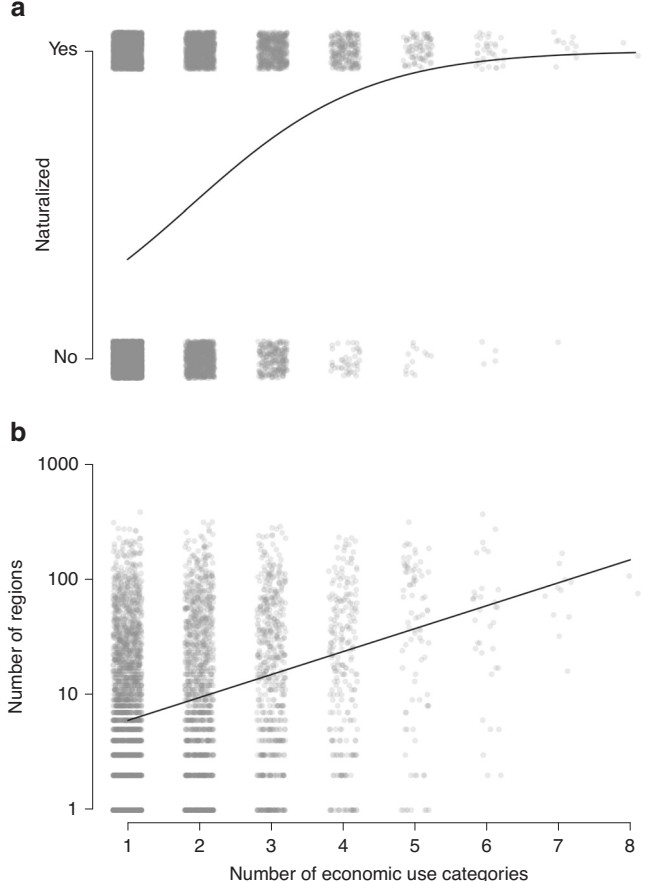

**Fig. 3 Relationship between the number of economic uses and naturalization success. a** Naturalization incidence of economic plants. **b** Naturalization extent (i.e., the number of GloNAF regions) for those economic plants that have become naturalized. The data points have been jittered to increase visibility. The fitted lines are (**a**) from a binomial GLM, and (**b**) from a Kendall–Theil Sen Siegel nonparametric regression.

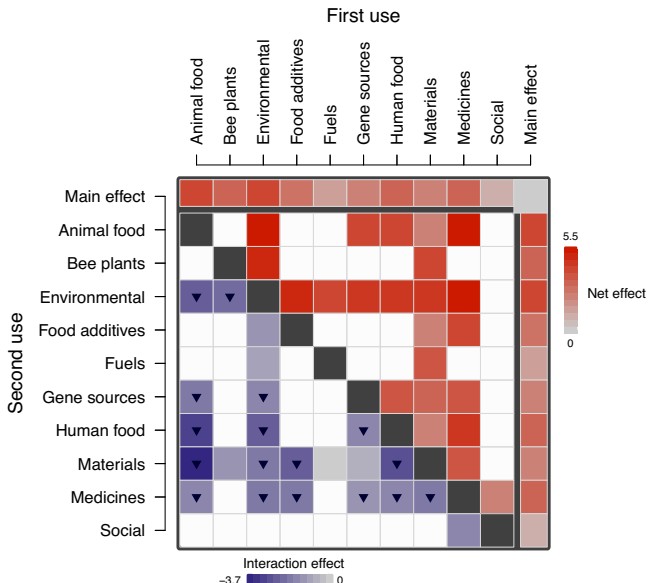

**Fig. 4 Naturalization success of taxa with different combinations of two economic uses.** Redish and bluish fill colors of the squares correspond to the magnitude of the coefficients from binomial GLMs for each pairwise combination of two economic uses (see Supplementary Table 3). The top row and right outermost column correspond to the main effects of each economic use, which, with the exception of fuels and social, were all significantly positive (all $p < 0.001$) compared with taxa with no economic use. The colors of the squares below the diagonal indicate the magnitude of the coefficients of the interaction term between both economic uses, which were all negative. Significant ($p < 0.05$) interaction coefficients are indicated with a black downward pointing triangle. Significance of the coefficients was determined with two-sided z-tests without correction for multiple comparisons. The colors of the squares above the diagonal (not including the top row and the right outermost column) indicate the magnitude of the net effect of the main and interactive effects of the two economic uses. Main effects for invertebrate food and non-vertebrate poisons are not shown, because each of their combinations with other economic use classes included fewer than ten taxa, resulting in high uncertainties of the estimates or no data. For the same reason, combinations of two economic uses that included fewer than ten taxa are also not shown (i.e., white squares).

interaction terms in the GLM relating naturalization success to each combination of two economic uses (Fig. 4 and Supplementary Table 3) indicate that the effects of the single uses are not fully additive. In other words, there are diminishing returns of having more than one economic use for naturalization success. The results are very similar when we use a binomial GLM that did not correct for phylogeny (Supplementary Table 4).

**Geographic patterns in naturalization of economic plants.** Compared with continental species richness (using the Taxonomic Databases Working Group (TDWG) continent scheme), the numbers of economic plants are higher than expected among the native floras of Africa, temperate Asia, Australasia, Europe, and North America (from Mexico north), and lower than expected for the Pacific Islands and South America (including the Central American countries south of Mexico; Fig. 5a). If one accounts for the fact that some taxa are native to multiple continents by analyzing the subset of economic plants native to only a single continent, the number of economically exploited taxa is higher than expected only among the native floras of temperate Asia, Europe, and North America (Fig. 5b). In other words, northern temperate regions contribute disproportionately to the global pool of economic plants.

The naturalized flora of each TDWG continent includes economic plant taxa originating from nearly every other continent, except Antarctica (Fig. 6). However, most intracontinental and intercontinental flows of naturalized economic plants are either larger or smaller than the values expected based on the sizes of the pools of economic plants from the respective donor continents (Fig. 6 and Supplementary Fig. 6). This indicates differential naturalization success among plants originating from the different continents. Temperate Asia is disproportionally overrepresented as a donor of naturalized economic taxa in all regions except the Pacific Islands (Fig. 6 and Supplementary Fig. 6). Tropical Asia is also overrepresented as a donor of naturalized economic plants to most continents, the exceptions being Europe (where tropical Asian plants are significantly underrepresented), Australasia, and North America (Fig. 6 and Supplementary Fig. 6). Other than the two Asian regions, both Africa and Europe are notable for having larger than expected intracontinental flows of naturalized economic taxa and for contributing disproportionately to the naturalized economic flora of Antarctica (Fig. 6 and Supplementary Fig. 6). The remaining continents are in most cases underrepresented as donors of naturalized economic plants (Fig. 6 and Supplementary Fig. 6). So, overall, economic plants originating from Asia, particularly temperate Asia, have the highest likelihood of naturalization.

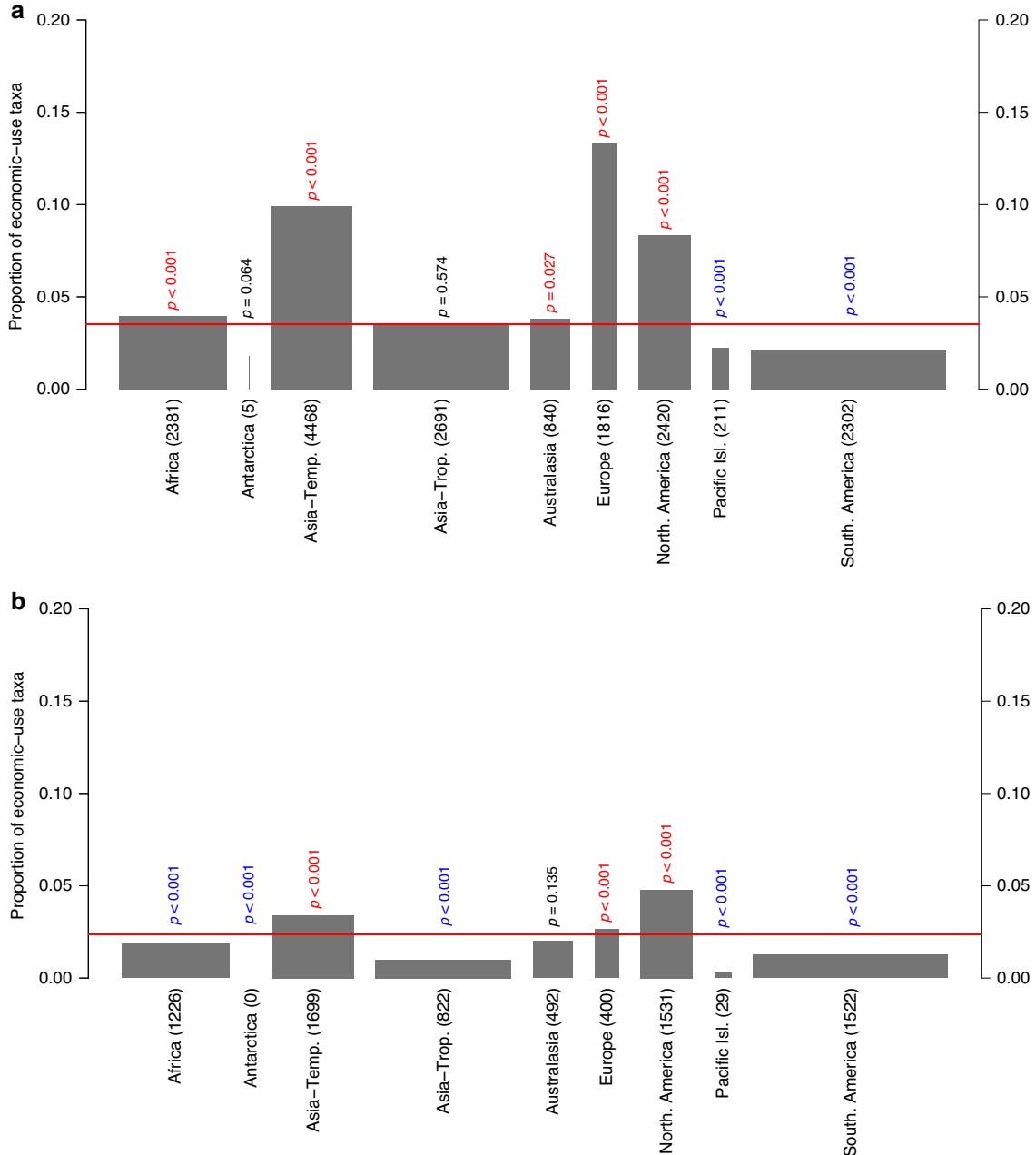

**Fig. 5 Estimated proportions of native plants that have an economic use for each TDWG continent.** Spineplots for (**a**) the taxa in the WEP dataset ($n = $ 11,485; excluding taxa only known from cultivation) and (**b**) the subset of taxa with a single native TDWG continent ($n = 7721$). The width of each bar is proportional to the number of taxa in each regional flora. The red horizontal lines indicate the proportion of taxa in the global flora that occur (**a**) in the WEP dataset (i.e., have an economic use) or (**b**) the respective subset of WEP taxa with a single native TDWG continent. $P$ values from two-sided resampling tests indicate whether the proportion of naturalized taxa is significantly higher (red) or lower (blue) than expected or does not deviate from expectations (black). No corrections were made for multiple comparisons. The number of taxa with an economic use from each continent is indicated in brackets.

While 39.9% of the naturalized taxa in GloNAF has a reported economic use (Fig. 1), this global percentage is lower than the percentage of economic taxa within regional naturalized floras in 848 of the 861 GloNAF regions (median percentage of economic plants in regional naturalized floras is 70.2%; Fig. 7, Supplementary Fig. 7, and Supplementary Data 1). This pattern indicates that naturalized economic plants occur in many more regions than naturalized noneconomic plants. In other words, many regional naturalized floras share the same economic plants. The percentage of economic plants in naturalized floras showed a clear latitudinal trend, increasing sharply from c. 50% at high

latitudes to c. 75% at the equator (binomial GLM: $t = -12.88$, $p <$ 0.001; Fig. 7). However, percentages were indistinguishable between island and mainland regions (island main effect: $t =$ 0.70, p = 0.485; island–latitude interaction: $t = -0.86$, $p = 0.388$; Fig. 7). The dominant use in most regions is either environmental or medicinal (Supplementary Fig. 8), reflecting the large number of taxa cultivated for these uses, and, in the case of environmental taxa, an increased likelihood of naturalization.

**Phylogenetic structure in economic use and naturalization.** Economic plants come from many different clades of the global

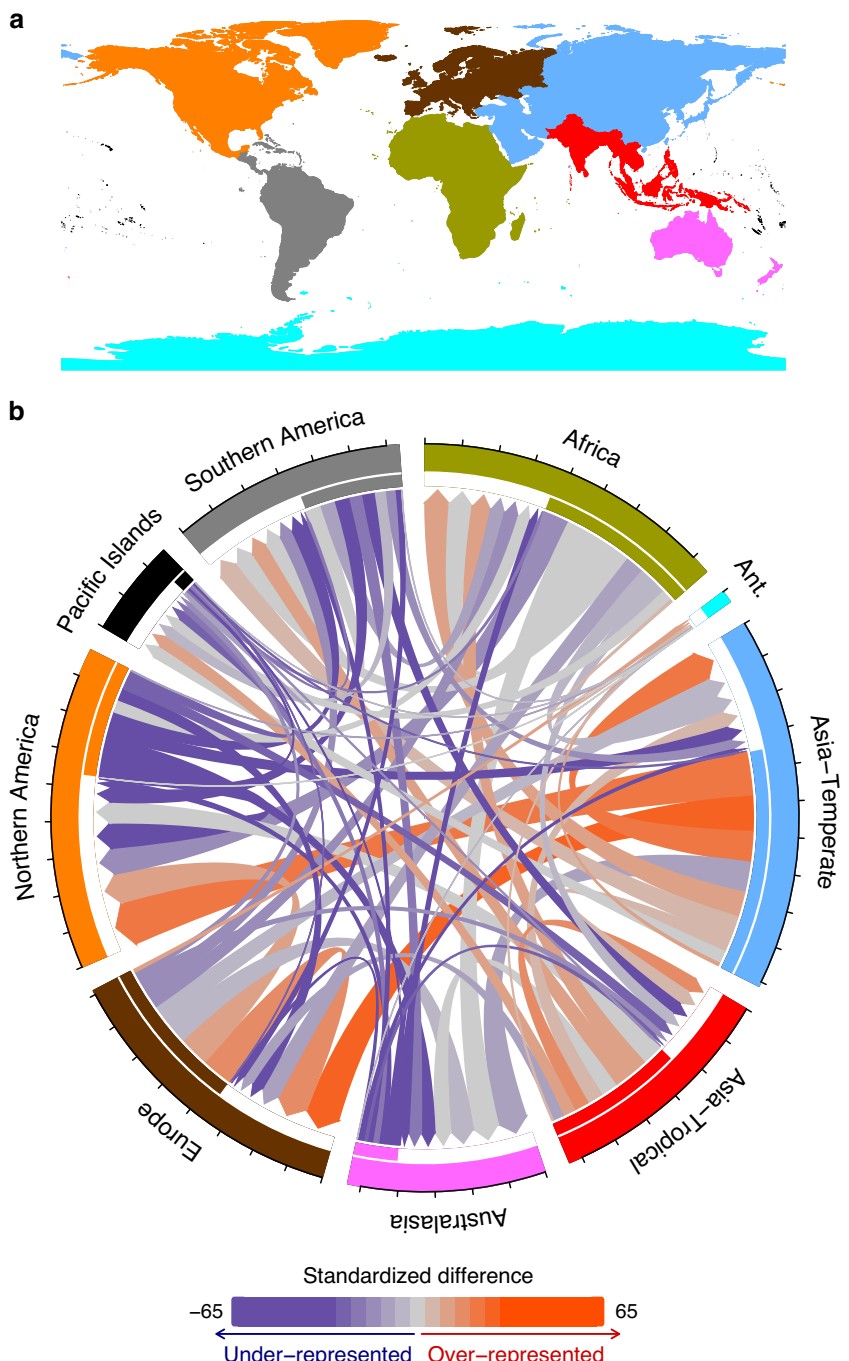

**Fig. 6 Flows of naturalized economic plants among the TDWG continents. a** Map of the TDWG continents. **b** Flow diagram in which the outer circle shows the cumulative number of naturalized economic plants that a continent has donated to other continents (or to itself) and received from those continents. Each tick interval along the outer circle corresponds to 1000 taxa. The colors of the flow arrows indicate whether the flows are larger (red color tones) or smaller (blue color tones) than expected based on random sampling (see Supplementary Fig. 6 for which flows are significantly larger or smaller than expected).

seed plant phylogeny, but our resampling test showed that they are still phylogenetically clustered (the standardized effect size (SES) = −37.07, $p < 0.001$; Fig. 8). Moreover, among the economic plants, the taxa within each of the economic use categories are also significantly phylogenetically clustered (Supplementary Figs. 9 and 10).

The taxa comprising the global naturalized flora also represent a significantly clustered subset of the global seed plant phylogeny (SES = −33.49, $p < 0.001$; Fig. 8), a pattern that is partly caused by the high proportion of economic plants that have become naturalized and by the phylogenetic clustering of these taxa (Fig. 8).

Nevertheless, after accounting for the phylogenetic clustering of economic plants, the naturalized taxa also still show significant phylogenetic clustering (SES = −25.72, $p < 0.001$; Fig. 8). In other words, the phylogenetic bias with regard to economic uses of plants partly explains the phylogenetic pattern in naturalized plants.

## Discussion
By combining the largest currently available databases on economic uses and naturalization success, we show how the varied

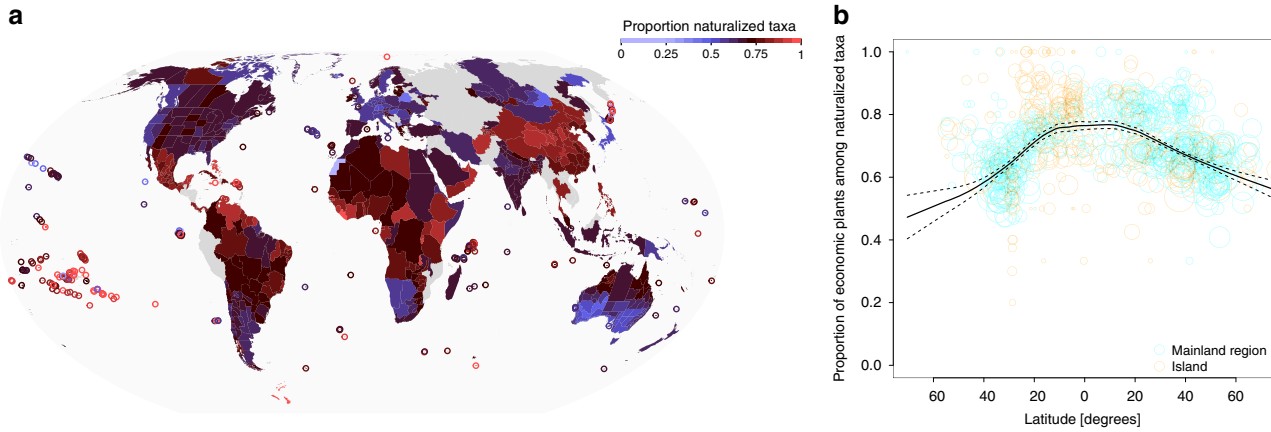

**Fig. 7 The proportion of economic plants in regional naturalized alien floras. a** Map of the world showing for each of the GloNAF regions the proportion of naturalized taxa that have an economic use. Circles were used to increase the visibility of small islands and island groups on the map. **b** Relationship between latitude and proportion of economic plants in regional naturalized floras. Cyan and orange circles correspond to mainland regions (n = 542) and islands (n = 319), respectively. The size of the circles is proportional to the natural logarithm of the number of naturalized alien plants in a region. The black solid line and the dashed lines are the fitted LOESS smoother (with the default span = 0.5) and its 95% confidence interval from a generalized additive model (R package gam[66]).

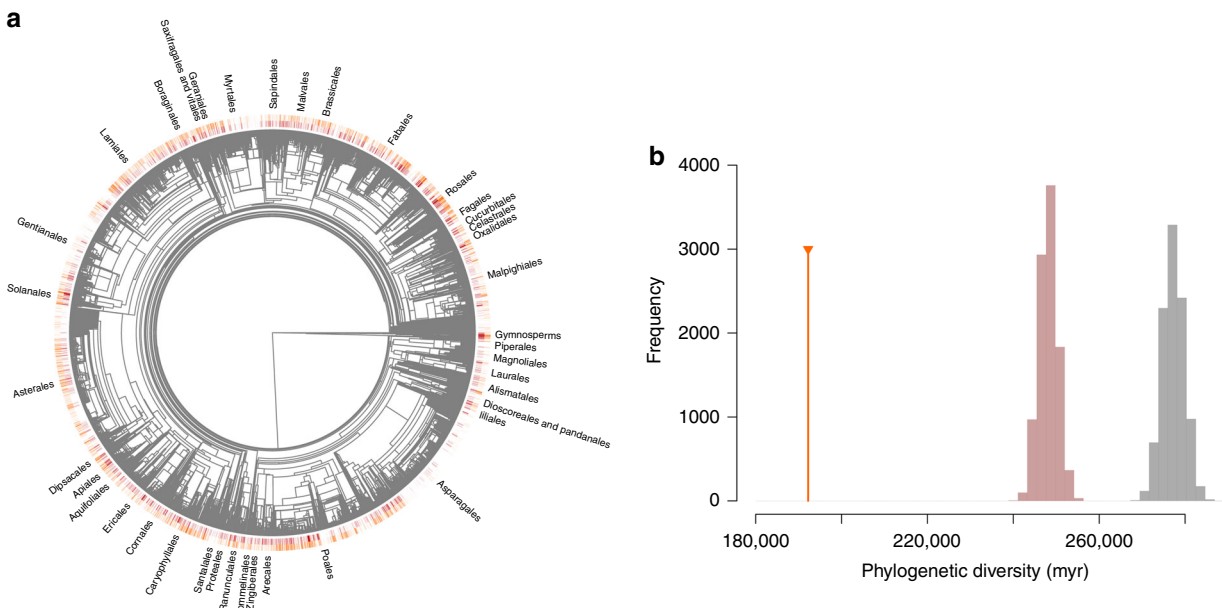

**Fig. 8 Phylogenetic structure of economic use and naturalization success. a** Phylogenetic tree of the extant global seed plant flora (n = 326,101) showing the distribution of economic plants in red, and the distribution of naturalized taxa in orange at the tips of the tree. Gymnosperms and the names of the angiosperm orders with >500 taxa are provided for orientation. **b** Test for the role of economic use in the phylogenetic structure for naturalization success. The orange line shows the phylogenetic diversity (in million years, myr) encompassed by the full sample of naturalized plants. The gray histogram is the distribution of phylogenetic diversity expected among naturalized plants if they would be randomly drawn from the extant global flora. The red histogram is the distribution of phylogenetic diversity expected among naturalized plants if they would be randomly drawn from the extant global flora with the restriction that the proportion of naturalized taxa with economic use(s) remains the same as the observed proportion. A smaller than expected phylogenetic diversity indicates phylogenetic clustering.

uses of plants by humans contribute to their ability to establish outside of their native ranges. The percentage of naturalized taxa among economic plants is almost 18 times higher than among the global seed plant flora without economic uses (41.0 vs. 2.3%, Fig. 1), indicating that intentionally introduced alien plants are highly likely to become naturalized. Furthermore, we show that the likelihood of naturalization increases with the number of economic uses. Our analyses further show that naturalization success varies among economic uses, with taxa grown as animal food or for environmental uses, contributing disproportionately

more to the global naturalized flora. For species with multiple economic uses, the specific combination of uses also mattered, and particularly if a focal economic use was combined with an environmental one, the likelihood of naturalization was always high. While taxa originating from the Northern Hemisphere are generally overrepresented in the global pool of economically used plants, those from temperate and tropical Asia are most likely to be naturalized in other continents. The percentage of economic plants in regional naturalized floras declines with latitude. Finally, we showed that the phylogenetic structure in

naturalization success is partly, but not fully, caused by a phylogenetic bias in the plants that have been selected by humans for economic use. Altogether, our results point to a strong role of economic use of plants in driving global plant naturalization patterns.

The large proportion of economic plants that have become naturalized strongly suggests that cultivation for economic use is the major pathway for the introduction of naturalized alien plants in regions across the globe. In the global naturalized flora, 39.9% of the taxa are known to have an economic use. Strikingly, however, this percentage is substantially higher in the vast majority (98.5%) of the regional naturalized floras, with a median of 70.2%. This deviation indicates that many regions around the world share the same naturalized plants with economic uses, and thus that they differ more in the naturalized plants without economic uses. In line with this, the median number of regions in which economic plants have become naturalized exceeds the median number of regions in which plants without economic uses have become naturalized (8 vs. 3 regions; Supplementary Fig. 2). An explanation could be that most of the naturalized plants without economic uses have been introduced accidentally (e.g., as contaminants of imported grain, wool, or fibers), a process that is likely more stochastic and spatially sporadic than the directed and repeated introduction of economic plants. Overall, these findings indicate that economic plants disproportionately contribute to global biotic homogenization.

All 12 categories of economic uses have proportions of naturalized taxa that by far exceed the percentage of the global flora that has naturalized (3.7%). For bee plants and plants used for the production of non-vertebrate poisons, over 70% of the taxa have become naturalized, and for another eight categories, the percentage is over 50%. The proportion of naturalized taxa is the lowest for gene sources used for genetic improvement of crops (27.3%), most likely because these taxa are not cultivated in large numbers. The high proportion of naturalized taxa in some of the economic use categories might be explained by most of their taxa having additional economic uses. For bee plants and non-vertebrate poisons, more than 90% of the taxa have an additional economic use, whereas for gene sources this is the case in less than 30% of the taxa. So, it could be that some economic use categories have high percentages of naturalized plants because their taxa frequently have multiple uses. When we focused on the taxa that have only a single economic use—to get an unbiased estimate of their contributions to naturalization success—then the categories animal food and environmental use had the highest proportions of naturalized taxa. Animal food plants are used as fodder or forage (Supplementary Fig. 11), which are usually grown in large quantities, and repeatedly, to feed the large global populations of livestock[47]. Environmental plants include predominantly taxa used as ornamentals (Supplementary Fig. 11), which are likely to be grown in many gardens around the world. The other, smaller subcategories of environmental uses, such as agroforestry, erosion control, and soil improvers, have even larger proportions of naturalized taxa than the ornamental subcategory (Supplementary Fig. 11), most likely because those plants are frequently directly planted into seminatural landscapes. So, plants in the animal food and environmental categories have contributed particularly many taxa to the naturalized floras, most likely because they are widely and repeatedly cultivated in large numbers, and thus benefit from a high propagule pressure.

Both the likelihood of naturalization and the number of regions, in which an economic plant is naturalized, increased with the number of economic uses. A plausible explanation for this is that planting frequency, and thus propagule pressure, is higher when a plant has multiple uses. A similar result was reported in a study analyzing the factors that determine the naturalization of

Central European species in North America—species that were utilized for a wider range of purposes by humans were more likely to naturalize[27]. This effect was manifested via the effect on residence time as species with a broader range of economic uses were likely to be introduced earlier in history, and thus had more opportunities to naturalize[27,48]. It has also been shown that species with multiple introduction pathways are more likely to have negative impacts[49]. So, the more uses a plant has, the more likely it is that it will naturalize.

In addition, we found that it also matters which combination of multiple economic uses a plant has. Among the taxa with two economic uses, the likelihood of naturalization was in most cases largely increased when an economic use is combined with an environmental one, and decreased when combined with the use as material. This provides further support for our conclusion that environmental plants and material plants have high and low naturalization probabilities, respectively. Most plant taxa in the subcategories of the category materials, such as the largest subcategory wax, are not cultivated very widely, and were possibly selected for specific qualities rather than for a high yield. The only exceptions were plants in the subcategories fiber and wood, which are planted more widely, and have relatively high likelihoods of naturalization (Supplementary Fig. 11). Having two (or more) economic uses resulted in a higher probability of naturalization overall, and the magnitude was mainly determined by additive effects of the economic uses. However, there were also diminishing returns of having more than one economic use, and the magnitude and significance of these interactions depended on the specific combination of economic uses (Fig. 4). So, although the likelihood of naturalization increases with the number of economic uses, the specific combination of the uses matters.

Relative to the size of the native floras, taxa from the Northern Hemisphere (temperate Asia, Europe, and North America) and Africa were overrepresented among the economic plants. Given the overall high likelihood of naturalization of economic plants, this pattern of disproportional overrepresentation could explain why the Northern Hemisphere continents are overrepresented as donors of naturalized alien plants generally[39]. In other words, Northern Hemisphere plants might not have a higher innate ability to naturalize, but are simply more likely to have been introduced elsewhere for economic use. If one only considers the economic plants that continents have donated, temperate Asia, as well as tropical Asia, is still overrepresented as a donor of naturalized plants in most continents, suggesting that many Asian plants have a high innate naturalization potential. However, Africa, Europe, and North America are then actually underrepresented instead of overrepresented in the naturalized floras of most other continents. This strongly suggests that the overrepresentation of European and North American plants in naturalized floras globally[39] is more likely due to their higher likelihood of being introduced elsewhere in the world (as economic plants) than due to a higher innate naturalization potential of plants from those continents. The reasons for differences in innate naturalization potential of taxa from different continents and how this potential might change in the future, due to, e.g., climate change, require more research.

The proportion of economic plants in regional naturalized floras is generally higher than the proportion in the global naturalized flora, but also varies among regions. Island floras usually have larger proportions of naturalized and invasive aliens than mainland regions[37,38]. While this pattern may reflect a reduced resistance against introduced aliens within ecologically simple and competitively naïve island communities[50,51], we had also expected that it could partly be driven by the need of humans to import alien plants into relatively species-poor island systems in order to meet economic needs. However, the proportion of

economic plants in naturalized island floras was not higher than in naturalized mainland floras. Apparently, in addition to the intentionally introduced economic plants, most islands also receive many unintentional introductions of alien species. We furthermore showed that the proportion of economic plants in the naturalized floras increased toward the equator. In other words, tropical regions usually have higher proportions of economic plants in their naturalized floras than temperate regions. A potential explanation could be that the species-rich tropical regions, which have stronger biotic interactions[52], have a higher resistance against invasion by alien organisms[53], and that a constantly high propagule pressure and specific traits allowed some of the economic plants to overcome such strong resistance. Alternatively, it could be that unintentionally transported species (i.e., noneconomic plants) are less likely to be introduced to many tropical regions due to a less strong integration into global trade networks[44].

Among the extant global seed plant flora, we found a strong phylogenetic structure in naturalization success. This is in line with the results of a previous study showing that certain plant families (e.g., Poaceae) are strongly overrepresented in the GloNAF, and other families (e.g., Orchidaceae) are strongly underrepresented[37]. This suggests that certain taxa in certain clades have higher naturalization potential than others. However, the phylogenetic bias in naturalization success may partly be caused by phylogenetic bias in the introduction of alien taxa. Indeed, among the extant global seed plant flora, we also found a strong phylogenetic structure for economic use of plants, and among the economic plants for most of the individual economic use categories. This indicates that closely related taxa are likely to share characteristics that make them suitable for economic use, or that they have similar histories of being brought into cultivation. When we corrected our test of phylogenetic structure in naturalization success for the economic use of plants, we indeed found that the phylogenetic structure, although it remained highly significant, became weaker. This shows that at least part of the phylogenetic patterns in naturalization success can be explained by phylogenetic biases with regard to the plants that humans use economically.

In conclusion, our global-scale analysis provides strong evidence that economic use in general, as well as the number and nature of economic uses, has strong effects on naturalization success. In particular, plants grown for animal food or environmental purposes (the latter represented mainly by ornamentals), which are most likely to be cultivated widely and in large numbers, have the highest likelihood of naturalization. Changes in the extent of cultivation may thus also affect future naturalization patterns. For example, the increasing cultivation of biofuel crops[54] may increase the naturalization success of plants in this economic use category.

While it has been suggested that European plants are successful invaders in many other parts of the world due to their innate invasiveness[42], our results suggest that their success is largely, though not exclusively, driven by their overrepresentation among economic plants. On the other hand, plants from temperate Asia are disproportionally overrepresented among economic plants, and also appear to have an innate greater naturalization potential. Furthermore, although the high naturalization success of some phylogenetic clades is likely to reflect innate naturalization properties of the species, our results indicate that phylogenetic biases in the economic use of plants partly underlie the phylogenetic pattern in naturalization success. Taken together, our results suggest that to fully unravel the drivers of naturalization success of plants, we must take into account their unique relationships with humans, and specifically how they are used in global economic activities. With ongoing climate change, it is

likely that in many regions there will be a turn over in the economic plants that can be grown outdoors, as well as in the demands for certain plants (e.g., for biofuels). The resulting changes in the cultivation of economic plants are likely to determine future naturalization patterns.

## Methods
**Databases used**. To assess how economic uses contribute to global naturalization success of plants, we combined data from several databases. Our global dataset for the economic uses of plants was extracted from the WEP database (National Plant Germplasm System GRIN-GLOBAL; https://npgsweb.ars-grin.gov/gringlobal/taxon/taxonomysearcheco.aspx, Accessed 7 Jan 2016), which is based on a book by Wiersema and León[45]. The WEP database provides information on 16 categories of economic uses (Table 1) for >15,000 vascular plant taxa (i.e., species, subspecies, and varieties) globally. The economic categories also have subcategories, ranging from one subcategory for the category "bee plants" to 25 for "gene sources" (106 subcategories in total; also see Supplementary Fig. 11). Four of the main categories, however, are not true economic use categories, as they refer to plants that either have negative impacts (harmful organism hosts, vertebrate poisons, and weeds) or are included in the Convention on International Trade in Endangered Species of Wild Fauna and Flora. These four categories were therefore not considered in our analyses, which reduced the dataset to 11,878 taxa. We also excluded the very few fern taxa (due to limited phylogenetic information), so that the final WEP dataset consisted of 11,685 taxa of seed plants (*Spermatophyta*). Although the WEP database is the largest database on economic plants, it does not claim to be fully comprehensive[45]. Indeed, according to online garden-plant encyclopedia, there are another c. 60,000 plant species that are also grown, at least occasionally, in domestic gardens[19]. Nevertheless, the WEP database is likely to include most of the economic plants that are widely cultivated.

To assess the naturalization success of all taxa in the WEP dataset, we used the GloNAF database[46]. This database contains lists of naturalized vascular plant taxa for 861 regions (countries or subnational administrative units), ranging in size from 0.03 to 6,864,961 km$^2$ (median size is 15,152 km$^2$) and covering >80% of the terrestrial ice-free surface globally. The GloNAF database includes a total of 13,083 taxa that according to the original data sources are naturalized, meaning that they have established self-sustaining wild populations[55]. In addition to information on their naturalization status, we also compiled for each taxon in the WEP dataset information on their continents of origin. For this, we considered their native status in each of the nine major biogeographically defined continental regions recognized by the Biodiversity Information Standards (also known as the TDWG[56]). These TDWG continents largely correspond to the geographical continents, but Asia has been split into temperate Asia and tropical Asia, and the Pacific Islands constitute a single TDWG continent (Fig. 6a). Most of these native range data were extracted from the same online database from which we also extracted the WEP data (i.e., National Plant Germplasm System GRIN-GLOBAL). However, for 139 taxa without native range information in that database, we extracted native range data from other internet sources (in particular from the World Checklist of Selected Plant Families (WCSP); http://wcsp.science.kew.org/). Ultimately, we found data on the TDWG continents of origin for 11,485 of the 11,685 economic plants in the WEP dataset. The remaining 200 taxa are known from cultivation only, are novel hybrid taxa, or have an unknown origin.

To be able to combine the different datasets, we first harmonized the taxonomic names of all taxa in each dataset according to The Plant List (http://www.theplantlist.org/), using the package Taxonstand[57] in R[58]. We only kept taxa with accepted names in The Plant List.

**Phylogenetic trees**. To be able to account for phylogenetic nonindependence of taxa in the statistical analyses, and to assess phylogenetic biases among plants with economic uses and among naturalized plants, we constructed a phylogenetic tree for the global flora (all 326,101 taxa with accepted names present in The Plant List). As basis for this tree, we used the phylogenetic tree constructed by Smith and Brown[59], which is *hitherto* the most comprehensive tree for seed plants. We first standardized the taxon names of this tree according to The Plant List using the package Taxonstand[57], and pruned the tree to only include taxa with accepted names in The Plant List. Some plants with accepted names in The Plant List were not found in the tree, and we therefore added them manually to the tree to construct the global flora tree using the R package phytools[60]. Specifically, 20.47% of taxa were missing from the tree but had congeneric species in the tree, so we added them to the root of that genus. When the whole genus was missing from the tree (1.34% of taxa), we added them to the family root. We then constructed the WEP tree by pruning the global flora tree to only include the taxa found in the WEP database.

**Statistical analyses**. All statistical analyses were done using R version 3.6.1[58].

To test whether the percentage of naturalized taxa in the WEP dataset (i.e., among plants with economic uses) is significantly higher than expected relative to the global seed plant flora, we used resampling tests. We randomly drew a number of taxa equal to the number of taxa in the WEP dataset ($n = 11,685$) from the

global seed plant flora ($n = 326,101$ according to The Plant List, Accessed 1 Oct 2019), and repeated this process 9999 times. Then we compared the observed number of naturalized plants among the WEP taxa (i.e., economic plants) to the distribution of the numbers of naturalized species resulting from the random samples. If the observed number is in the upper or lower 2.5% quantiles of the resampled values, we consider it to be significantly higher or lower, respectively, than expected. As alternative tests, we also ran a binomial GLM and a phylogenetic binomial GLM. The latter was done to account for phylogenetic nonindependence of taxa and was run using the phyloglm and phylolm functions of the "phylolm" R package[61].

To test whether the percentage of naturalized taxa is higher or lower than expected for each of the individual economic use categories relative to the entire WEP dataset, we again used a resampling test. We randomly drew a number of taxa equal to the number of naturalized taxa in the WEP dataset from the WEP dataset, and repeated this 9999 times. Then we compared for each economic use category the observed number of naturalized taxa to the distributions resulting from the random samples. We did this for the entire WEP dataset, and, to avoid confounding due to some taxa having multiple economic uses, we also did this for the subset of taxa with a single economic use. As alternative tests, we again also ran a binomial GLM and a phylogenetic GLM.

To test whether the naturalization success of taxa in the WEP database increased with the number of economic uses, we did two tests. First, we ran a binomial GLM and a phylogenetic GLM of naturalization status (yes, no) on the number of economic use categories per taxon. Second, for the subset of taxa that were naturalized, we did a Kendall–Theil Sen Siegel nonparametric regression of the log10-transformed number of GloNAF regions on the number of economic use categories per taxon, implemented in the R package mblm[62]. In addition, to account for phylogenetic nonindependence of taxa in the log10-transformed number of GloNAF regions, we also ran a phylogenetic LM.

To test how combinations of two economic uses affected naturalization success, we ran a binomial GLM and a phylogenetic binomial GLM with a logit link for the taxa with no, one, or two economic uses (i.e., taxa with >2 uses were not considered in this analysis). We included each of the economic uses as main effects, and we included all possible two-way combinations of economic uses, but excluded combinations that did not exist in the WEP database.

To test whether there are biases in the continents of origin of economic plants and in the continents of origin of naturalized economic plants, we again used resampling tests. We assessed whether the contribution of each continental region to the global pool of economic plants is larger or smaller than expected based on the number of taxa native to the continent. To do this, we compared the observed number of economic taxa from each continent to the number based on 9999 random draws from the extant global flora equal to the total number of taxa in the WEP database. Since few data on the number of native plant taxa per TDWG continent exist, we estimated these numbers by extrapolation of the known native origins of 130,641 plant species in the WCSP database to the total number of 326,101 accepted taxa in The Plant List (see ref. 39 for a similar approach). The WCSP database does not include all plant families yet, and it could be geographically biased. However, previous studies showed that all 52 TDWG level-2 regions are well represented in the WCSP database[63], and that the extrapolated numbers of taxa per continent do not strongly diverge from previous estimates[39]. To assess whether the intracontinental and intercontinental flows of naturalized economic plants are larger or smaller than expected, we compared the observed flows to the ones based on 9999 random draws from the WEP dataset.

To test whether the proportion of economic plants among naturalized taxa in a region changes with latitude and differs between islands and mainland regions, we used a binomial GLM. The proportion of economic plants was the response variable, and absolute latitude and whether the region is an island or part of the mainland, as well as their interaction, were the explanatory variables. We accounted for overdispersion by using the quasi-binomial setting, which uses a dispersion parameter to correct the standard errors of the estimates of the binomial model[64].

To test whether there is phylogenetic structure in economic use and naturalization success, we used resampling tests. First, we calculated Faith's phylogenetic diversity[65] encompassed by the total sample of economic plants, and for the total sample of naturalized plants. Then we randomly sampled from the phylogenetic tree of the extant global seed plant flora, the same numbers of taxa as in the economic plant dataset and in the naturalized flora dataset, respectively. This was done 9999 times to get the distributions of expected values. If the observed phylogenetic diversity is in or below the lower 2.5% quantile of the resampled values, there is significant phylogenetic structure. As an index of the strength of the phylogenetic structure, we calculated the SES as the difference between the observed phylogenetic diversity and the mean value of the distribution of expected values divided by the standard deviation of the expected values. To test whether a potential phylogenetic structure in economic use underlies the potential phylogenetic structure in naturalization success, we did a second resampling test. We again randomly sampled the number of naturalized taxa from the extant global flora, but with the restriction that the proportion of resampled taxa that had an economic use was the same as the observed proportion.

**Reporting summary**. Further information on research design is available in the Nature Research Reporting Summary linked to this article.

## Data availability
The databases that we used are all publicly available, and the references are provided in the "Methods." The data (csv files) that support the findings of this study are available in figshare with the identifier https://doi.org/10.6084/m9.figshare.12278057.v1.

## Code availability
The R code used for the statistical analyses is available in figshare with the identifier https://doi.org/10.6084/m9.figshare.12278057.v1.

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

## Acknowledgements

We thank Vanessa Pasqualetto for extracting the data from the World Economic Plants database. M.v.K. thanks the German Research Foundation DFG for funding (grants 264740629 and 432253815). X.X. thanks Fudan's Undergraduate Research Opportunities Program for support and the National Top Talent Undergraduate Training Program for funding of the internship. P.P. and J.P. were supported by EXPRO grant No. 19-28807×(Czech Science Foundation) and long-term research development project RVO 67985939 (The Czech Academy of Sciences). F.E., B.L., and D.M. appreciate funding by the Austrian Science Foundation FWF (grant I2086-B16).

## Author contributions

M.v.K. devised the initial idea, contributed to data analysis, and led the writing. X.X. combined the datasets and contributed to data analysis and writing. Q.Y., N.M., Z.Z., and T.S.F. contributed ideas and contributed to data analysis and writing. W.D., F.E., H.K., J.P., P.P., P.W., D.M., and B.L. contributed data and contributed to writing.

## Competing interests

The authors declare no competing interests.
