## [Peer Review File · Nature Communications]

Peer Review File - Reviewers' comments, first round:

Reviewer #1 (Remarks to the Author):

Comments to authors

van Kleunen and colleagues examine and describe links between the economic use of plants and their probability of becoming naturalised at a global scale. The study is innovative, impressively comprehensive and very well presented. I imagine the paper will be very well received by researchers and practitioners around the world. It seems highly suitable for Nature Comms, attracting what I imagine will be many citations.

I thought the paper and study were excellent; globally important and interesting, carefully considered and contextualised, thoughtful and innovative, well explained and justified. My comments are accordingly minor; they are mostly intended to improve presentation and make a few more bits of information more readily available to readers (thus increasing likely number of citations).

Minors comments

abstract: it would be helpful here to briefly state what is meant by "economic plants"; very clear later on, but a small note here will help to ensure the paper is very accessible and readers know exactly what it is about from the get-go.

Abstract: also suggest you ensure meaning of "percentage of economic plants" is clearer; I was unsure exactly what this referred to on first reading. How about something like: "The % of regional naturalised floras that are made up of economic plants"

P4L75: different tense used in first and second parts of sentence. Change "are more" to "could be" or equivalent

P4L79: similar issue of clarity as noted above. Change to something like: "The proportion of species in regional floras that are naturalised..."

P5L103: for example?

P5L108-109: meaning unclear

P7L150 (and possibly earlier): I was a little thrown by "environmental" economic plants initially as name is not terribly intuitive, so suggest on first introduction of this term you provide a little more context/info (I note Table 1, but brief explanation in main text would still be helpful).

P7L153: insert "only" before "used" to ensure very clear

P7L165-168: not essential, but this sentence was a bit of hard work, so would be great if it would be rephrased slightly for clarity.

P8L14: "expected based on...???"

P9L194: it would be nice to pick this up again in the discussion; do you expect this to change in the future? What might be implications of that?

P9L201: "shared among regional floras": meaning not immediately obvious. Suggest edit for clarity

P9L207-208: if general trend, why start sentence talking about islands? Also, would be helpful to give some numbers to illustrate degree of difference, e.g. "such that tropical islands/regions have xxx" (suggestion only)

P10L219-220: it would be helpful to either talk about phylogenetic clustering OR phylogenetic diversity. I appreciate that they tell you the same thing (albeit in different directions) but there is a lot going on in the paper, so making it as easy as possible for readers to follow would be good (i.e. suggest retain focus on clustering so people don't have to do mental gymnastics when reading these sentences)

P10L221-222: is there a direction of causality issue here? If economic use is a driver of trends in naturalisation surely it is phylogenetic bias in the former that (at least partially) drives phylogenetic bias in the latter? As written, it sounds like you're suggesting the other way around. Perhaps you meant to write "partly explains" rather than "is partly explained [by]...??"

P11L248: I know you've got Fig 7 but a table in the SI that breaks down regions of the world and provides these stats (e.g. % of naturalised flora that are economic plants, plus % that are attributed to various economic use categories and subcategories) would be incredibly useful to researchers and practitioners, especially at regional levels. I imagine that kind of info would

generate many citations and help to inform regional-level responses to invasion and biosecurity. Discussion: I found the Discussion a little repetitive of the Results section, but the extra interpretation was useful. If looking to cut words, it might be possible to edit the results to reduce repetition.

P16L378: a couple of extra sentences that discuss the future implications of this work be a welcome addition to the paper. Given changes in trade routes, and increases in trade more generally, I would expect that some of the trends observed to date might change. Touching on that briefly would be a nice way to finish a paper and no doubt be instructive for readers (including policy makers) of the paper.

P20L482: any reason for the WCSP database to be biased? I would assume that some regions of the world would be better known than others, thus distorting the data a bit? What might be the implications of such bias on your findings if it were to exist?

Methods: generally very well explained and contextualised. Impressive effort!

Fig 2: it would be good to extend the y axes as differences among categories are currently very hard to see.

Fig 2 & similar figs: I found the grey bars a bit distracting, and don't think they add much. Can you get rid of them?

Fig 4: why include the "no use" categories twice? The row and column seem to show same data, as would be expected (as if no use, impossible to have a second use).

Fig 6: explanation for what outer most circle shows is needed.

Fig 7a: hard to discern differences in this fig. How about using e.g. blues for regions where they are less than the average % across all regions and reds for regions that are more than the average? Something like that would help to highlight regional differences more and show hotspots and coldspots.

Fig S5: what are the regions? Is there an average size? Would be helpful info for those unfamiliar with GloNAF (e.g. short explanation in figure caption)

Reviewer #2 (Remarks to the Author):

The manuscript entitled "Economic use of plants is key to unravelling naturalization success" is based on an evaluation of economic use of plants as driver of their naturalization success. The authors have merged two impressive databases, one of economic use of plants and the other of naturalized plants around the world, to find that plants with economic use are more likely to become naturalized. This type of solid evidence was needed to test this pattern, that has been suggested before. Of particular importance is that through their analyses the authors show that the economic use of plants is behind the naturalization success of plant species that are native to the Northern Hemisphere, and not their inherent ability to become naturalized (as was believed before). However, there are two issues with the manuscript that may need to be clarified:

- The database of naturalized alien plants the authors are using (GloNAF) may be biased in a way that confuses these results. The naturalized species lists used to build GloNAF include, in many cases, species that have been seen to grow outside of cultivation. For example, many important crops, such as wheat (*Triticum aestivum*), corn (*Zea mays*), rice (*Oryza sativa*), soybean (*Glycine max*) and sunflower (*Helianthus annuus*) are frequently seen growing on their own mainly because a proportion of the seeds from the cultivated area falls to the soil during the harvest and these seeds germinate the following growing season. However, this would only be one generation of plants growing unassisted by humans and cannot be considered as a naturalized population yet. The problem is that annual crops are sown every year with the same species within the same farmland. Therefore, how can we know if a population of an annual plant species has established a self-sustaining population if that same plant species is sown every year and every year a proportion of the seeds of the harvest go into the soil? More than 50% of the plant species included in GloNAF are annual plants (Pysek et al. 2017). From these, the species that have an

economic use are possibly sown with a very high frequency on the same area. As a result, many of these plant species may have been classified as naturalized just because every year plants can be seen outside of cultivation, although they may just be there because of the seeds that were lost from the previous harvest (and not because they constitute a self-sustaining population). To sum up, there may be a cause-effect confusion here: annual species naturalization success increases with economic use because the same plant species is planted frequently on a large scale (i.e. high propagule pressure)? Or is it that annual species with economic use are more likely to be classified as naturalized just because every year seeds are not harvested completely from the cultivated area and the next growing season plants grow outside cultivation? Going back to the example of highly common annual crops, wheat, corn, rice, soybean and sunflower are all among the 10% most successful species naturalizing outside their native range, according to GloNAF (van Kleunen et al. 2019). Unfortunately, it is hard to tell if these plant species have really become naturalized in all the recorded regions or if it is just spontaneous regeneration after each harvest. This is not an issue with perennial plants, because these are usually planted once and left for many years to grow. If these perennial populations produce new recruits that eventually become reproductive this would be evidence of a self-sustaining population. To evaluate if annual plants are introducing an important bias to these results one possibility is to redo the analyses without considering annual plants and check if the results change.

- Invasive and naturalized plants are not the same. Only a small proportion of naturalized plants become invasive. Therefore, the factors that drive naturalization success may differ substantially from the factors that drive invasion success. The authors should not claim invasive results when they are looking at naturalized plants. This occurs all through the manuscript.

Further comments:

Why is the number of species included in GloNAF in the results section 11,976 and in Figure 1 12,010?

Line 713: "compared to taxa with no economic use"

Line 321-324: Nice contribution

References

van Kleunen, M., P. Pyšek, W. Dawson, F. Essl, H. Kreft, J. Pergl, P. Weigelt, A. Stein, S. Dullinger, C. König, B. Lenzner, N. Maurel, D. Moser, H. Seebens, J. Kartesz, M. Nishino, A. Aleksanyan, M. Ansong, L. A. Antonova, J. F. Barcelona, S. W. Breckle, G. Brundu, F. J. Cabezas, D. Cárdenas, J. Cárdenas-Toro, N. Castaño, E. Chacón, C. Chatelain, B. Conn, M. de Sá Dechoum, J.-M. Dufour-Dror, A. L. Ebel, E. Figueiredo, O. Fragman-Sapir, N. Fuentes, Q. J. Groom, L. Henderson, Inderjit, N. Jogan, P. Krestov, A. Kupriyanov, S. Masciadri, J. Meerman, O. Morozova, D. Nickrent, A. Nowak, A. Patzelt, P. B. Pelsner, W.-s. Shu, J. Thomas, A. Uludag, M. Velayos, A. Verkhosina, J. L. Villaseñor, E. Weber, J. J. Wieringa, A. Yazlık, A. Zeddiam, E. Zykova, and M. Winter. 2019. The Global Naturalized Alien Flora (GloNAF) database. *Ecology* 100:e02542.

Pyšek, P., J. Pergl, F. Essl, B. Lenzner, W. Dawson, H. Kreft, P. Weigelt, M. Winter, J. Kartesz, and M. Nishino. 2017. Naturalized alien flora of the world. *Preslia*. 89:203-274.

Reviewer #3 (Remarks to the Author):

van Kleunen and coauthors present an analysis of naturalised plant species across the globe, and show that species used for economic purposes are overrepresented. This observation is both profoundly unsurprising and important. Plants with economic uses have presumable been transported widely by people, and this 'propagule' pressure likely explains the strong association between naturalisation and economic use. However, the magnitude of this effect is seemingly large, and the findings presented here suggest that any study looking to understand patterns of naturalisation in plants should first account for whether they have economic uses before appealing to ecological or evolutionary explanations. The authors additionally suggest that phylogenetic patterning in plants with economic uses might explain the phylogenetic patterning in naturalisation

(although I suspect the true explanation might be somewhat more complicated, see comments below).

Overall, I enjoyed reading this paper, and the Discussion was excellent. I thought it was a generally fun analysis, perhaps revealing an underappreciated truism. However, I found the statistical approach simplistic and lacking in rigour. Although I suspect the results will prove robust, it is the authors' responsibility to demonstrate this.

General Comments

1. The authors extract information on naturalised plants from the GloNAF database. I am not familiar with the criteria for inclusion in this database, but certainly a definition of 'naturalisation' is critical. It would, for example, be awkward if agricultural usage was sufficient for inclusion in the GloNAF database.

2. Much of the authors' statistical testing was based on a null of random resampling. For example, lines 443-452 (see also lines 453-460):

"To test whether the percentage of naturalized taxa in the WEP dataset (i.e. among plants with economic uses) is significantly higher than expected relative to the global seed plant flora, we used resampling tests. We randomly drew a number of taxa equal to the number of taxa in the WEP dataset (n=11,685) from the global seed plant flora ..."

This is problematic because species in the WEP database are not a random sample of taxa, but represent a phylogenetically structured subset of taxa. Thus it is inappropriate to simulate the structure of the WEP database by randomly drawing taxa from the global taxon pool. This problem permeates the authors' resampling tests. There are methods by which the authors could resample taxa so as to retain the phylogenetic structure of the original dataset; however, a more powerful approach would be to use a phylogenetic binomial regression, for example, with naturalised as the response and economic uses as the predictor (see comment below).

3. Lines 461-466. Similarly, the authors use two approaches to test whether naturalisation success increases with number of economic uses, yet neither corrects for phylogenetic non-independence. Given the stated phylogenetic structure in both attributes this is an important omission. A phylogenetic binomial regression and pgl analysis could easily address this gap (see R packages *phyloIm* and *caper*). The same criticism applies to the following analyses described on lines 467-473.

4. The exploration of continental biases in the origins of economic and naturalised plants (lines 474-485) could similarly be addressed using generalised linear models and examining the residuals, but this is perhaps less critical.

Minor comments

1. Lines 109-111. The authors suggest phylogenetic patterning in economic uses could explain phylogenetic patterning in naturalisation; however, it is probable that the traits associated with economic uses (e.g. the ability to grow under a wide range of conditions) also facilitate naturalisation, and thus the link is through plant traits rather than direct.

2. Lines 114-121. The authors outline a series of rather descriptive questions to motivate their study, e.g. "Are there biases in ...", and present no falsifiable hypotheses. It would have been better to state and then test specific hypotheses.

3. Line 456. The authors refer to "this dataset" without clarifying which dataset they are referring to.

4. Lines 493-508. Phylogenetic signal is usually measured with respect to a model of Brownian motion (e.g. using Blomberg's K or Pagel's Lambda). Nonetheless, I think the metric used here (standard effect size of PD) is fine, but perhaps should be referred to a measure of 'phylogenetic structure' rather than 'phylogenetic signal' so as to avoid confusion.

Response Letter

Below we give point-by-point responses to the all reviewer comments. Our responses are in blue. Please, note that the line numbers refer to the clean version of the manuscript without track changes.

Reviewers' comments:

Reviewer #1 (Remarks to the Author):

Comments to authors

van Kleunen and colleagues examine and describe links between the economic use of plants and their probability of becoming naturalised at a global scale. The study is innovative, impressively comprehensive and very well presented. I imagine the paper will be very well received by researchers and practitioners around the world. It seems highly suitable for Nature Comms, attracting what I imagine will be many citations.

I thought the paper and study were excellent; globally important and interesting, carefully considered and contextualised, thoughtful and innovative, well explained and justified. My comments are accordingly minor; they are mostly intended to improve presentation and make a few more bits of information more readily available to readers (thus increasing likely number of citations).

Response: We thank the reviewer for this very positive evaluation.

Minors comments

abstract: it would be helpful here to briefly state what is meant by “economic plants”; very clear later on, but a small note here will help to ensure the paper is very accessible and readers know exactly what it is about from the get-go.

Response: We added a brief explanation of “economic plants” to the abstract (line 27).

Abstract: also suggest you ensure meaning of “percentage of economic plants” is clearer; I was unsure exactly what this referred to on first reading. How about something like: “The % of regional naturalised floras that are made up of economic plants”

Response: We now write “In regional naturalized floras, the percentage of economic plants” (lines 34-35).

P4L75: different tense used in first and second parts of sentence. Change “are more” to “could be” or equivalent

Response: We corrected this (line 76).

P4L79: similar issue of clarity as noted above. Change to something like: “The proportion of species in regional floras that are naturalised...”

Response: We changed this into “The proportion of naturalized species is usually lower in tropical than in temperate regional floras” (lines 80-81).

P5L103: for example?

Response: We now added examples of families that have more naturalized species than expected (lines 105-106).

P5L108-109: meaning unclear

Response: As this sentence was not essential, we deleted it.

P7L150 (and possibly earlier): I was a little thrown by “environmental” economic plants initially as name is not terribly intuitive, so suggest on first introduction of this term you provide a little more context/info (I note Table 1, but brief explanation in main text would still be helpful).

Response: We now provide more context on “environmental” economic plants at first mention (lines 155-156).

P7L153: insert “only” before “used” to ensure very clear

Response: We changed this accordingly (line 158).

P7L165-168: not essential, but this sentence was a bit of hard work, so would be great if it would be rephrased slightly for clarity.

Response: We rephrased this sentence as “While the effects of multiple economic uses mainly follow from the main effects of the single uses, the negative interaction terms in the GLM relating naturalization success to each combination of two economic uses (Fig. 4, Supplementary Table 2) indicate that the effects of the single uses are not fully additive. In other words, there are diminishing returns of having more than one economic use for naturalization success.” (lines 174-178).

P8L184: “expected based on...???”

Response: We now clarify this (line 195).

P9L194: it would be nice to pick this up again in the discussion; do you expect this to change in the future? What might be implications of that?

Response: It is difficult to speculate on what might cause differences in innate invasion potential of species from different continents and whether this might change in the future. However, we now mention this as a potential future research topic (lines 339-341).

P9L201: “shared among regional floras”: meaning not immediately obvious. Suggest edit for clarity

Response: To improve clarity, we rephrased this as “This pattern indicates that naturalized economic plants occur in many more regions than naturalized non-economic plants. In other words, many regional naturalized floras share the same economic plants.” (lines 213-215).

P9L207-208: if general trend, why start sentence talking about islands? Also, would be helpful to give some numbers to illustrate degree of difference, e.g. “such that tropical islands/regions have

xxx” (suggestion only)

Response: We are not entirely sure why the reviewer asks why we start the sentence talking about islands, as we only mention islands in the second part of the sentence. However, we now rephrased the sentence, and hope this addressed the point. In addition, we now mention that the proportion of economic plants in naturalized floras increased from c. 0.5 at high latitudes to c. 0.75 at the equator (lines 215-219).

P10L219-220: it would be helpful to either talk about phylogenetic clustering OR phylogenetic diversity. I appreciate that they tell you the same thing (albeit in different directions) but there is a lot going on in the paper, so making it as easy as possible for readers to follow would be good (i.e. suggest retain focus on clustering so people don't have to do mental gymnastics when reading these sentences)

Response: We analysed phylogenetic clustering by comparing the observed and the expected phylogenetic diversity of all economic plants, and of all naturalized plants. In the sentence pointed out by the reviewer, we tried to clarify this. However, we realized that this might indeed require some mental gymnastics, and we therefore now only use “phylogenetic clustering” in the main text of the Results and Discussion. As phylogenetic clustering was measured by comparing the observed and the expected phylogenetic diversity of the naturalized taxa (or the economic use plants), we explain this in the Methods (lines 526-528), as well as in the caption of Fig. 8.

P10L221-222: is there a direction of causality issue here? If economic use is a driver of trends in naturalisation surely it is phylogenetic bias in the former that (at least partially) drives phylogenetic bias in the latter? As written, it sounds like you're suggesting the other way around. Perhaps you meant to write “partly explains” rather than “is partly explained [by]..”??

Response: That is a good point. We indeed meant “partly explains”, and changed this accordingly (line 236).

P11L248: I know you've got Fig 7 but a table in the SI that breaks down regions of the world and provides these stats (e.g. % of naturalised flora that are economic plants, plus % that are attributed to various economic use categories and subcategories) would be incredibly useful to researchers and practitioners, especially at regional levels. I imagine that kind of info would generate many citations and help to inform regional-level responses to invasion and biosecurity.

Response: We have now added a supplementary table (Supplementary Table 4) where we provide for each region the number of naturalized taxa, the number of those that are economic plants, and the number in each economic use category. We did not include the economic use subcategories, as there are 106 subcategories in total, which would make the table very large.

Discussion: I found the Discussion a little repetitive of the Results section, but the extra interpretation was useful. If looking to cut words, it might be possible to edit the results to reduce repetition.

Response: We have tried to remove some of the repetition, but did not change the Discussion too much as Reviewer 3 thought it was excellent as it stood.

P16L378: a couple of extra sentences that discuss the future implications of this work be a welcome addition to the paper. Given changes in trade routes, and increases in trade more generally, I would expect that some of the trends observed to date might change. Touching on

that briefly would be a nice way to finish and paper and no doubt be instructive for readers (including policy makers) of the paper.

Response: We added a few sentences on the future implications (lines 395-399).

P20L482: any reason for the WCSP database to be biased? I would assume that some regions of the world would be better known than others, thus distorting the data a bit? What might be the implications of such bias on your findings if it were to exist?

Response: We cannot fully exclude a distortion, but if there is one we believe it to be small. We used the same approach of estimating the numbers of species per continent in van Kleunen et al. (2015). There we showed that the numbers of species estimated for each continent are quite similar to estimates given in other studies. We now mention this in the Methods (lines 512-514).

Methods: generally very well explained and contextualised. Impressive effort!

Response: We thank the reviewer for this compliment.

Fig 2: it would be good to extend the y axes as differences among categories are currently very hard to see.

Response: We now extended the y-axes (Fig. 2, and also in Fig. 5).

Fig 2 & similar figs: I found the grey bars a bit distracting, and don't think they add much. Can you get rid of them?

Response: We removed the grey bars (Fig. 2, Fig. 5, Supplementary Fig. 11).

Fig 4: why include the "no use" categories twice? The row and column seem to show same data, as would be expected (as if no use, impossible to have a second use).

Response: We included it as a reference category, and we included it twice so that it is easier to compare the colors in the rows and the columns. However, we realized that the term "no use" might be confusing. Therefore, we renamed it "main effect", and now mention in the caption that it is meant as a reference category (Fig. 4).

Fig 6: explanation for what outer most circle shows is needed.

Response: We added an explanation (Fig. 6).

Fig 7a: hard to discern differences in this fig. How about using e.g. blues for regions where they are less than the average % across all regions and reds for regions that are more than the average? Something like that would help to highlight regional differences more and show hotspots and coldspots.

Response: We changed the color scheme to make it easier to discern the regions that have values higher and lower than the median percentage (Fig. 7).

Fig S5: what are the regions? Is there an average size? Would be helpful info for those unfamiliar with GloNAF (e.g. short explanation in figure caption)

Response: We now added more information on the GloNAF regions and their sizes in the Methods section (lines 425-428).

Reviewer #2 (Remarks to the Author):

The manuscript entitled “Economic use of plants is key to unravelling naturalization success” is based on an evaluation of economic use of plants as driver of their naturalization success. The authors have merged two impressive databases, one of economic use of plants and the other of naturalized plants around the world, to find that plants with economic use are more likely to become naturalized. This type of solid evidence was needed to test this pattern, that has been suggested before. Of particular importance is that through their analyses the authors show that the economic use of plants is behind the naturalization success of plant species that are native to the Northern Hemisphere, and not their inherent ability to become naturalized (as was believed before). However, there are two issues with the manuscript that may need to be clarified:

Response: We thank the reviewer for the overall positive evaluation.

- The database of naturalized alien plants the authors are using (GloNAF) may be biased in a way that confuses these results. The naturalized species lists used to build GloNAF include, in many cases, species that have been seen to grow outside of cultivation. For example, many important crops, such as wheat (*Triticum aestivum*), corn (*Zea mays*), rice (*Oryza sativa*), soybean (*Glycine max*) and sunflower (*Helianthus annuus*) are frequently seen growing on their own mainly because a proportion of the seeds from the cultivated area falls to the soil during the harvest and these seeds germinate the following growing season. However, this would only be one generation of plants growing unassisted by humans and cannot be considered as a naturalized population yet. The problem is that annual crops are sown every year with the same species within the same farmland. Therefore, how can we know if a population of an annual plant species has established a self-sustaining population if that same plant species is sown every year and every year a proportion of the seeds of the harvest go into the soil? More than 50% of the plant species included in GloNAF are annual plants (Pysek et al. 2017). From these, the species that have an economic use are possibly sown with a very high frequency on the same area. As a result, many of these plant species may have been classified as naturalized just because every year plants can be seen outside of cultivation, although they may just be there because of the seeds that were lost from the previous harvest (and not because they constitute a self-sustaining population). To sum up, there may be a cause-effect confusion here: annual species naturalization success increases with economic use because the same plant species is planted frequently on a large scale (i.e. high propagule pressure)? Or is it that annual species with economic use are more likely to be classified as naturalized just because every year seeds are not harvested completely from the cultivated area and the next growing season plants grow outside cultivation? Going back to the example of highly common annual crops, wheat, corn, rice, soybean and sunflower are all among the 10% most successful species naturalizing outside their native range, according to GloNAF (van Kleunen et al. 2019). Unfortunately, it is hard to tell if these plant species have really become naturalized in all the recorded regions or if it is just spontaneous regeneration after each harvest. This is not an issue with perennial plants, because these are usually planted once and left for many years to grow. If these perennial populations produce new recruits that eventually become reproductive this would be evidence of a self-sustaining population. To evaluate if annual plants are introducing an important bias to these results one possibility is to redo the analyses without considering annual plants and check if the results change.

Response: The GloNAF database is a compilation of many different sources that each lists introduced alien plants for a specific region or for multiple regions. When a source is not clear on whether a species is naturalized or not, the species is classified as “alien” for that region in GloNAF. In that case, the alien species could be naturalized, but it could also be a casual (i.e. regularly escaping but not establishing long-lasting populations) or just an alien species that is cultivated. However, when a source explicitly states that a species is naturalized, the species is classified as “naturalized” for that region in GloNAF. For the analyses presented here, we only included GloNAF data for species classified as naturalized species. We now state this explicitly, and also provide a definition of naturalized (lines 428-430).

Although, in most regions, annual crops found occasionally in the wild are classified as casuals, we cannot fully exclude the possibility that some of the data sources used for the compilation of the GloNAF database incorrectly classified some annual crop species as naturalized. Such an incorrect classification should be equally likely for perennial economic plants. Populations of perennial species may persist for a long time, even after a field or garden where they were cultivated has been abandoned. Even if annual and perennial economic plants have in some cases been incorrectly classified as naturalized, the 18 times higher naturalization probability that we found for economic than for non-economic plants would only disappear if most naturalized species have been incorrectly classified as such. Therefore, we are convinced that the concern of the reviewer does not warrant that we redo all analyses after exclusion of annual plants. Note that we could not quickly do the requested re-analysis as we do not have information on growth form for all 11,686 economic plant taxa. Furthermore, it should be noted that the examples of annual plants mentioned by the reviewer are widely grown crop species, which are only a relatively small selection of the many economic plant species. Although some of those species mentioned by the reviewer have multiple uses, they all are grown for “human food”, and this category does not stand out as having a higher naturalization success compared to the other categories (see Fig. 2b). Finally, the reviewer’s concern is partly based on the reviewer’s statement that “More than 50% of the plant species included in GloNAF are annual plants (Pyšek et al. 2017)”. This statement, however, is incorrect. Pyšek et al. (2017) write that among the top 200 of most widely naturalized species, 45% are annuals. In the entire GloNAF database of >13,000 taxa, only c. 20% are annuals. In conclusion, we respectfully believe that the reviewer’s concern is not justified.

- Invasive and naturalized plants are not the same. Only a small proportion of naturalized plants become invasive. Therefore, the factors that drive naturalization success may differ substantially from the factors that drive invasion success. The authors should not claim invasive results when they are looking at naturalized plants. This occurs all through the manuscript.

Response: We agree and are fully aware that naturalized and invasive alien species are not the same, and we do not believe that we wrote anywhere in the MS that naturalized and invasive species are the same. We used “invasive plants” several times when we referred to studies on invasive plants. We also used several times “invasion success”, but this is not the same as “invasive species” as invasion success refers to the full sequence of invasions from introduction to being invasive (sensu Blackburn et al. 2011, A proposed unified framework for biological invasions, TREE 27, 333-339). Both naturalization and invasion are stages of the invasion process, and thus “naturalization incidence” and “naturalization extent” are components of invasion success. However, to avoid confusion, we carefully checked our wording and changed it

whenever we thought it could be interpreted as suggesting that we looked at invasive species. Specifically, we replaced “invasion success” with “naturalization success” throughout the manuscript.

Further comments:

Why is the number of species included in GloNAF in the results section 11,976 and in Figure 1 12,010?

Response: We thank the reviewer for spotting this discrepancy. The number should be 12,013, and we corrected this accordingly (line 131, Fig. 1). Note, that the numbers in the current version deviate slightly from the ones in the previous submission, as we found some taxa had been incorrectly matched.

Line 713: “compared to taxa with no economic use”

Response: We corrected this accordingly (line 755).

Line 321-324: Nice contribution

Response: We thank the reviewer for the comments and general support of the manuscript.

References

van Kleunen, M., P. Pyšek, W. Dawson, F. Essl, H. Kreft, J. Pergl, P. Weigelt, A. Stein, S. Dullinger, C. König, B. Lenzner, N. Maurel, D. Moser, H. Seebens, J. Kartesz, M. Nishino, A. Aleksanyan, M. Ansong, L. A. Antonova, J. F. Barcelona, S. W. Breckle, G. Brundu, F. J. Cabezas, D. Cárdenas, J. Cárdenas-Toro, N. Castaño, E. Chacón, C. Chatelain, B. Conn, M. de Sá Dechoum, J.-M. Dufour-Dror, A. L. Ebel, E. Figueiredo, O. Fragman-Sapir, N. Fuentes, Q. J. Groom, L. Henderson, Inderjit, N. Jogan, P. Krestov, A. Kupriyanov, S. Masciadri, J. Meerman, O. Morozova, D. Nickrent, A. Nowak, A. Patzelt, P. B. Pelsler, W.-s. Shu, J. Thomas, A. Uludag, M. Velayos, A. Verkhosina, J. L. Villaseñor, E. Weber, J. J. Wieringa, A. Yazlık, A. Zeddani, E. Zykova, and M. Winter. 2019. The Global Naturalized Alien Flora (GloNAF) database. *Ecology* 100:e02542.

Pyšek, P., J. Pergl, F. Essl, B. Lenzner, W. Dawson, H. Kreft, P. Weigelt, M. Winter, J. Kartesz, and M. Nishino. 2017. Naturalized alien flora of the world. *Preslia*. 89:203-274.

Reviewer #3 (Remarks to the Author):

van Kleunen and coauthors present an analysis of naturalised plant species across the globe, and show that species used for economic purposes are overrepresented. This observation is both profoundly unsurprising and important. Plants with economic uses have presumably been transported widely by people, and this ‘propagule’ pressure likely explains the strong association between naturalisation and economic use. However, the magnitude of this effect is seemingly large, and the findings presented here suggest that any study looking to understand patterns of naturalisation in plants should first account for whether they have economic uses before appealing to ecological or evolutionary explanations. The authors additionally suggest that

phylogenetic patterning in plants with economic uses might explain the phylogenetic patterning in naturalisation (although I suspect the true explanation might be somewhat more complicated, see comments below).

Overall, I enjoyed reading this paper, and the Discussion was excellent. I thought it was a generally fun analysis, perhaps revealing an underappreciated truism. However, I found the statistical approach simplistic and lacking in rigour. Although I suspect the results will prove robust, it is the authors' responsibility to demonstrate this.

Response: We thank the reviewer for this overall positive evaluation.

General Comments

1. The authors extract information on naturalised plants from the GloNAF database. I am not familiar with the criteria for inclusion in this database, but certainly a definition of 'naturalisation' is critical. It would, for example, be awkward if agricultural usage was sufficient for inclusion in the GloNAF database.

Response: We now state more clearly that we extracted only species from GloNAF that have been classified as naturalized by the original data sources, and we provide definition of naturalized (lines 428-430).

2. Much of the authors' statistical testing was based on a null of random resampling. For example, lines 443-452 (see also lines 453-460):

"To test whether the percentage of naturalized taxa in the WEP dataset (i.e. among plants with economic uses) is significantly higher than expected relative to the global seed plant flora, we used resampling tests. We randomly drew a number of taxa equal to the number of taxa in the WEP dataset (n=11,685) from the global seed plant flora ..."

This is problematic because species in the WEP database are not a random sample of taxa, but represent a phylogenetically structured subset of taxa. Thus it is inappropriate to simulate the structure of the WEP database by randomly drawing taxa from the global taxon pool. This problem permeates the authors' resampling tests. There are methods by which the authors could resample taxa so as to retain the phylogenetic structure of the original dataset; however, a more powerful approach would be to use a phylogenetic binomial regression, for example, with naturalised as the response and economic uses as the predictor (see comment below).

Response: We are convinced that resampling is a powerful and straightforward way to test whether certain groups of organisms are disproportionally over- or under-represented. The Reviewer is right that resampling does not account for any phylogenetic structure. However, neither of the characteristics that we are investigating, i.e. naturalization or economic use, are inherited directly by taxa from their ancestors and therefore do not follow the evolutionary processes assumed by phylogenetic analyses (e.g. these characteristics are not transitioning between states along the tree). Of course, as we discuss in the Introduction, the link between economic use and naturalization must be driven by some underlying mechanism; some of our suggested possibilities should act independently of phylogeny (e.g. economic use is associated with higher propagule pressure) others would potentially not (e.g. economic taxa are chosen for traits that also make them more likely to naturalize). While it will certainly be valuable to eventually disentangle the mechanisms, it is not within the scope of the current paper to answer all these questions. We would argue that even in an extreme case where the pattern is driven by one clade that is over-represented in the economic use dataset, it would still hold that the plants

we cultivate for economic uses are playing a major role in global naturalization patterns. So, one should take care that by doing a phylogenetic correction one does not throw out the baby with the bathwater. Nevertheless, to see how robust our findings are, we followed the advice of the reviewer to also use phylogenetic (binomial) regression. The conclusions did not change. We mention these additional analyses in the main manuscript (lines 134-135, 162-163, 178-179, 495-501) and present them in the supplements (Supplementary Tables 1 and 3, and Supplementary Figures 1 and 5).

3. Lines 461-466. Similarly, the authors use two approaches to test whether naturalisation success increases with number of economic uses, yet neither corrects for phylogenetic non-independence. Given the stated phylogenetic structure in both attributes this is an important omission. A phylogenetic binomial regression and pgl analysis could easily address this gap (see R packages `phylolm` and `caper`). The same criticism applies to the following analyses describe on lines 467-473.

Response: We now also did phylogenetic binomial regression and pgl for these analyses, and the conclusions did not change. We mention these additional analyses in the main manuscript and present them in the supplements (Supplementary Table 1).

4. The exploration of continental biases in the origins of economic and naturalised plants (lines 474-485) could similarly be addressed using generalised linear models and examining the residuals, but this is perhaps less critical.

Response: As the reviewer indicates that this complicated analysis may be less critical, and the other phylogenetic corrections did not change the conclusions, we refrained from doing such an analysis.

Minor comments

1. Lines 109-111. The authors suggest phylogenetic patterning in economic uses could explain phylogenetic patterning in naturalisation; however, it is probable that the traits associated with economic uses (e.g. the ability to grow under a wide range of conditions) also facilitate naturalisation, and thus the link is through plant traits rather than direct.

Response: We agree that this could be the case, and we now mention this (lines 110-112). However, even if it would not be due to the traits, a phylogenetic patterning in economic use could through the high propagule pressure associated with many economic plants cause a phylogenetic patterning in naturalization.

2. Lines 114-121. The authors outline a series of rather descriptive questions to motivate their study, e.g. “Are there biases in ...”, and present no falsifiable hypotheses. It would have been better to state and then test specific hypotheses.

Response: We have now replaced the four rather descriptive questions with five more specific questions that indicate our expectations/hypotheses (lines 116-124).

3. Line 456. The authors refer to “this dataset” without clarifying which dataset they are referring to.

Response: We now mention that it refers to the WEP dataset (line 479).

4. Lines 493-508. Phylogenetic signal is usually measured with respect to a model of Brownian motion (e.g. using Blomberg's K or Pagel's Lambda). Nonetheless, I think the metric used here (standard effect size of PD) is fine, but perhaps should be referred to as a measure of 'phylogenetic structure' rather than 'phylogenetic signal' so as to avoid confusion.

Response: We now replaced "phylogenetic signal" with "phylogenetic structure" throughout the manuscript.

REVIEWERS' COMMENTS, second round:

Reviewer #2 (Remarks to the Author):

The authors have addressed all my comments and the comments from the other reviewers. In particular I appreciate that the authors were open to constructive criticism and made many changes to include reviewers suggestions. With all the work done by the authors and the changes suggested by the reviewers I believe this manuscript is a valuable contribution to our understanding of biological invasions. I have no further comments.

Reviewer #3 (Remarks to the Author):

I appreciate the authors efforts to address concerns I raised on their previous submission. I was somewhat surprised, however, that the authors suggested that resampling was a powerful way to test whether certain groups of organisms are over- or under-represented. While their implementation is certainly simple, it is open to considerable bias, does not provide an estimate of the variance explained, and provides no mechanistic understanding. There is a large literature on constraining resampling that allows corrections for various biases (e.g. see publications by N. Gotelli), which the authors largely ignore. In contrast, regression approaches provide coefficient estimates and give an estimate of variance explained, correct for phylogenetic non-independence, and additionally allow for multiple predictors to be modelled simultaneously as well as their interaction terms.

The comment that phylogenetic non-independence is not problematic because neither naturalisation nor economic use are directly inherited suggests a misunderstanding of phylogenetic comparative methods (there is a very large literature on this subject that the authors can explore). The authors also suggest that even if patterns were driven by a single clade, their results indicating that "plants we cultivate for economic uses are playing a major role in global naturalization patterns" would still hold - but this is deeply misleading, and our inference would change importantly.

I recommend that the phylogenetic regression models be presented in the main text (the authors have already done all the hard work in fitting the models). I also suggest the authors publish the R code for their full analyses - why make it only available on 'reasonable request'?

This is an interesting and well-written manuscript - the authors have obviously done a large amount of work, and include many well presented figures - it deserves to be supported by rigorous and thoughtful analyses. This could be a great paper.

REVIEWERS' COMMENTS, third round -

Reviewer #2 (Remarks to the Author):

The authors have addressed all my comments and the comments from the other reviewers. In particular I appreciate that the authors were open to constructive criticism and made many changes to include reviewers suggestions. With all the work done by the authors and the changes suggested by the reviewers I believe this manuscript is a valuable contribution to our understanding of biological invasions. I have no further comments.

Response: We thank the reviewer for this positive evaluation.

Reviewer #3 (Remarks to the Author):

I appreciate the authors efforts to address concerns I raised on their previous submission. I was somewhat surprised, however, that the authors suggested that resampling was a powerful way to test whether certain groups of organisms are over- or under-represented. While their implementation is certainly simple, it is open to considerable bias, does not provide an estimate of the variance explained, and provides no mechanistic understanding. There is a large literature on constraining resampling that allows corrections for various biases (e.g. see publications by N. Gotelli), which the authors largely ignore. In contrast, regression approaches provide coefficient estimates and give an estimate of variance explained, correct for phylogenetic non-independence, and additionally allow for multiple predictors to be modelled simultaneously as well as their interaction terms.

Response: We agree that regression approaches have advantages. However, they also have the disadvantage that they require assumptions about the distribution of the data. Therefore, we do not believe that they are necessarily superior to re-sampling tests. As the re-sampling tests that we did are intuitively easy to understand, and because the figures (which have the benefit of showing the raw data on proportions of naturalized species in each category) are based on those tests, we strongly prefer to keep a focus on those tests in the main manuscript. We now, however, also increased the focus on results from the phylogenetic GLM analyses in the main text (see response below).

The comment that phylogenetic non-independence is not problematic because neither naturalisation nor economic use are directly inherited suggests a misunderstanding of phylogenetic comparative methods (there is a very large literature on this subject that the authors can explore). The authors also suggest that even if patterns were driven by a single clade, their results indicating that “plants we cultivate for economic uses are playing a major role in global naturalization patterns” would still hold - but this is deeply misleading, and our inference would change importantly.

Response: Overall, we agree with the reviewer that the phylogenetic regressions are valuable analyses for testing these relationships and in confirming the robustness of our results. We have therefore taken the reviewer’s suggestion to elevate results from these analyses to the main text (including adding a new table and updating the main figures). However, we would like to take the opportunity here to clarify our previous comments on the topic and to provide our reasoning for including both sets of analyses. We consider this manuscript as an early step in understanding

the role of the economic use of plants in global naturalization patterns and therefore believe that it is therefore valuable to simply demonstrate that the plants humans have cultivated economically are over-represented among naturalized species. We therefore prefer to keep our original analyses in the main text, which make intuitive comparisons of the proportion of economic plants that have become naturalized. We agree with the reviewer that understanding the mechanisms behind this association would be beneficial and we outline a number of potential mechanisms that we are not able to test with current data. Specifically we suggest that increased propagule pressure of economic species and/or preferential selection of certain traits in these species may increase their likelihood of naturalization. Our point in the initial response letter was that because each case of selection by humans for economic use and each case of naturalization are evolutionary independent events (i.e. the ‘switch’ to either status occurs at the tips and does not evolve along the tree as assumed in the evolutionary models underlying phylogenetic regression analyses), the observed association between the two cannot be caused directly by shared ancestry. Of course, there are mechanisms that could lead to phylogenetic signal in the relationship between economic use and naturalization. For example, closely related species may be more similar in traits that affect either their likelihood of being selected for economic use or of becoming naturalized. In this case, phylogenetically informed analyses may serve as an approximate means of controlling for non-independence among species in the inherited traits involved in either process, but this won’t necessarily help us to understand the mechanisms underlying the pattern. Take for example an extreme case where the association between economic use and naturalization is driven primarily by a single clade that is over-represented among the economic plants. A likely scenario is that species in this clade were selected so often for economic use because some characteristics common to the clade make them highly desirable for economic use. Three potential explanations for the observed link between economic use and naturalization could be: 1) this common desirable trait also makes the species more likely to naturalize (mechanistic link between economic use and naturalization); 2) this desirable trait also means that these species are grown more frequently, in more locations, and at higher abundances, increasing propagule pressure (mechanistic link between economic use and naturalization); 3) these species also share a different trait that is unrelated to economic use that increases their naturalization probability (no mechanistic link between economic use and naturalization). In this example, an analysis that controls for phylogenetic relationships among species may find no significant relationship between economic use and naturalization, even in the case of a true mechanistic link (e.g. possibility 1 or 2). This would be misleading, and publishing these results alone would potentially discourage further research that may lead to a better understanding of the underlying mechanism. In our case, the phylogenetic analyses arrive at nearly identical results as our original analyses, allowing us to confirm that a scenario similar to possibility 3 is not driving the patterns. However, for the reasons outlined above, in addition to the intuitive nature of our original analyses, we think that presenting both sets of analyses provides the most balanced view.

I recommend that the phylogenetic regression models be presented in the main text (the authors have already done all the hard work in fitting the models).

Response: We now put more emphasis on the phylogenetic GLMs. We have added the statistics that were previously in Supplementary Table 1 to the main text, and we added the results of a phylogenetic GLM to Figure 2 (previously this panel was part of Supplementary Figure 5). In addition, we now show in Figure 4 the coefficients of the phylogenetic GLM.

I also suggest the authors publish the R code for their full analyses – why make it only available on ‘reasonable request’?

Response: We now also publish the R code for the full analyses in figshare, and we adapted the Code Availability statement accordingly.

This is an interesting and well-written manuscript - the authors have obviously done a large amount of work, and include many well presented figures - it deserves to be supported by rigorous and thoughtful analyses. This could be a great paper.

Response: We thank the reviewer for this compliment.